# Phosphoprotein SAK1 is a regulator of acclimation to singlet oxygen in *Chlamydomonas reinhardtii*

**Setsuko Wakao[1], Brian L Chin[1†], Heidi K Ledford[1‡], Rachel M Dent[1§], David Casero[2¶], Matteo Pellegrini[2,3], Sabeeha S Merchant[3,4], Krishna K Niyogi[1,5,6]\***

[1]Department of Plant and Microbial Biology, University of California, Berkeley, Berkeley, United States; [2]Department of Molecular, Cell and Developmental Biology, University of California, Los Angeles, Los Angeles, United States; [3]Institute for Genomics and Proteomics, University of California, Los Angeles, Los Angeles, United States; [4]Department of Chemistry and Biochemistry, University of California, Los Angeles, Los Angeles, United States; [5]Howard Hughes Medical Institute, University of California, Berkeley, Berkeley, United States; [6]Physical Biosciences Division, Lawrence Berkeley National Laboratory, Berkeley, United States

**\*For correspondence:** niyogi@
berkeley.edu

**Present address:** [†]Department of Systems Biology, Harvard Medical School, Boston, United States; [‡]Nature News, Nature Publishing Group, Cambridge, United States; [§]School of Public Health, Univeristy of California, Berkeley, Berkeley, United States; [¶]Department of Pathology and Laboratory Medicine, University of California, Los Angeles, Los Angeles, United States

**Competing interests:** The authors declare that no competing interests exist.

**Reviewing editor**: Detlef Weigel, Max Planck Institute for Developmental Biology, Germany

**Abstract** Singlet oxygen is a highly toxic and inevitable byproduct of oxygenic photosynthesis. The unicellular green alga *Chlamydomonas reinhardtii* is capable of acclimating specifically to singlet oxygen stress, but the retrograde signaling pathway from the chloroplast to the nucleus mediating this response is unknown. Here we describe a mutant, *singlet oxygen acclimation knocked-out 1* (*sak1*), that lacks the acclimation response to singlet oxygen. Analysis of genome-wide changes in RNA abundance during acclimation to singlet oxygen revealed that SAK1 is a key regulator of the gene expression response during acclimation. The *SAK1* gene encodes an uncharacterized protein with a domain conserved among chlorophytes and present in some bZIP transcription factors. The SAK1 protein is located in the cytosol, and it is induced and phosphorylated upon exposure to singlet oxygen, suggesting that it is a critical intermediate component of the retrograde signal transduction pathway leading to singlet oxygen acclimation.

## Introduction

Growth of photosynthetic organisms depends on light energy, which in turn can cause oxidative damage to the cell if not managed properly (*Li et al., 2009*). Light intensity is highly dynamic in terrestrial and aquatic environments, and the cell must constantly control the dissipation of light energy to avoid photo-oxidative stress while maximizing productivity. In addition to being the site of photosynthesis, the chloroplast houses many essential biochemical reactions such as fatty acid and amino acid biosynthesis, but most of its proteins are encoded in the nucleus and must be imported after translation. Therefore the nucleus must monitor the status of the chloroplast and coordinate gene expression and synthesis of proteins to maintain healthy chloroplast functions.

It is known that signals originating from a stressed or dysfunctional chloroplast modulate nuclear gene expression, a process that is called retrograde signaling (*Nott et al., 2006*; *Chi et al., 2013*). In *Arabidopsis thaliana* the *gun* mutants have helped to define the field of chloroplast retrograde signaling, leading to the identification of GUN1, a pentatricopeptide repeat protein that is a regulator of this process (*Koussevitzky et al., 2007*), and pointing to the involvement of the tetrapyrrole biosynthetic pathway (*Vinti et al., 2000*; *Mochizuki et al., 2001*; *Larkin et al., 2003*; *Strand et al., 2003*; *Woodson and Chory, 2008*). A role for heme in retrograde signaling has been shown in *Chlamydomonas*

**eLife digest** Plants, algae and some bacteria use photosynthesis to extract energy from sunlight and to convert carbon dioxide into the sugars needed for growth. One by-product of photosynthesis is a highly toxic molecule called singlet oxygen. Typically, organisms deal with stressful events such as the presence of toxic molecules by producing new proteins. However, protein production is generally initiated in the nucleus of the cell, and photosynthesis is carried out in structures called chloroplasts. Cells must therefore be able to alert the nucleus to the presence of toxic levels of singlet oxygen in the chloroplasts.

Like some plants that can withstand a gradual decrease in temperature, but not a sudden cold snap, the alga *Chlamydomonas reinhardtii* is capable of resisting high doses of singlet oxygen if it has previously been exposed to low doses of the molecule. Wakao et al. exploited this ability to hunt for algae that are unable to acclimate to singlet oxygen, and found that these cells are unable to produce a protein called SAK1.

Wakao et al. reveal that many factors involved in the algae's cellular response to singlet oxygen depend on the presence of SAK1. In addition, the response of the algae cells to singlet oxygen differs to the one seen in the model plant *Arabidopsis thaliana*, suggesting that the two organisms have found different ways to deal with the same problem.

The location of a protein in a cell can give clues to its function. SAK1 is present in the fluid surrounding cellular compartments—the cytosol—which is consistent with it acting as a signaling molecule between the chloroplast and the nucleus. Wakao et al. present further evidence for this hypothesis by demonstrating that the number of phosphate groups attached on SAK1 changes when exposed to singlet oxygen—a feature often seen in signaling proteins. In addition, part of SAK1 resembles proteins that can bind to DNA, which indicates that SAK1 may be directly involved in initiating protein production.

The discovery of SAK1 represents a starting point for understanding how the site of photosynthesis, the chloroplast, communicates with the nucleus. It also has implications for developing plants and algae that have a higher tolerance to environmental stress conditions for agriculture and biofuel production.

reinhardtii as well (**von Gromoff et al., 2008**). Many of the *gun* studies were conducted in context of a dysfunctional chloroplast treated with norflurazon, an inhibitor of carotenoid biosynthesis. More recently a number of exciting advances have shed light on small molecules playing roles in retrograde stress signaling, including methylerythritol cyclodiphosphate, an intermediate of isoprenoid biosynthesis in the chloroplast (**Xiao et al., 2012**), 3-phosphoadenosine 5-phosphate (PAP) (**Estavillo et al., 2011**), as well as a chloroplast envelope transcription factor PTM (**Sun et al., 2011**). Plastid gene expression involving sigma factors has been implicated in affecting nuclear gene expression, although the mechanism is unknown (**Coll et al., 2009**; **Woodson et al., 2012**).

Activation of gene expression by reactive oxygen species (ROS) has been well documented (**Apel and Hirt, 2004**; **Mittler et al., 2004**; **Gadjev et al., 2006**; **Li et al., 2009**). Thus ROS have been proposed as a means for chloroplasts to signal stress to the nucleus and many examples of global gene expression changes in response to ROS have been described (**Desikan et al., 2001**; **Vandenabeele et al., 2004**; **Vanderauwera et al., 2005**). Singlet oxygen ($^1O_2$) is a highly toxic form of ROS that can be formed in all aerobic organisms through photosensitization reactions in which excitation energy is transferred from a pigment molecule to $O_2$. For example, porphyria in humans is caused by defects in tetrapyrrole metabolism that can lead to accumulation of photosensitizing intermediates, which generate $^1O_2$ in the light (**Straka et al., 1990**). In oxygenic photosynthetic organisms, $^1O_2$ is mainly generated at the reaction center of photosystem II, when triplet excited chlorophyll transfers energy to $O_2$ (**Krieger-Liszkay, 2005**). $^1O_2$ is the predominant cause of lipid oxidation during photo-oxidative stress (**Triantaphylidès et al., 2008**) and is associated with damage to the reaction center (**Trebst et al., 2002**). Because of the abundance and proximity of the two elements of $^1O_2$ generation, the photosensitizer chlorophyll and $O_2$, it was hypothesized that oxygenic photosynthetic organisms must have evolved robust means to cope with this ROS (**Knox and Dodge, 1985**). In *Arabidopsis*, the EX1 and EX2 proteins in the chloroplast are required for the execution of a $^1O_2$-dependent response: growth

arrest in plants and programmed cell death in seedlings, that is distinct from cell damage (*op den Camp et al., 2003*; *Wagner et al., 2004*; *Lee et al., 2007*). Different players in $^1O_2$ signaling have emerged recently, such as β-cyclocitral, an oxidation product of β-carotene in *Arabidopsis* (*Ramel et al., 2012*), a bZIP transcription factor (SOR1) responding to reactive electrophiles generated by $^1O_2$ (*Fischer et al., 2012*), and a cytosolic zinc finger protein conserved in *Arabidopsis* and *Chlamydomonas*, MBS (*Shao et al., 2013*). In the anoxygenic photosynthetic bacterium *Rhodobacter sphaeroides*, a σ$^E$ factor is responsible for the elicitation of the gene expression response to $^1O_2$ (*Anthony et al., 2005*).

The unicellular green alga *Chlamydomonas reinhardtii* is an excellent model organism for investigation of retrograde $^1O_2$ signaling. *Chlamydomonas* exhibits an acclimation response to $^1O_2$, in which exposure to a sublethal dose of $^1O_2$ leads to changes in nuclear gene expression that enable cells to resist a subsequent challenge with higher levels of $^1O_2$ (*Ledford et al., 2007*). We hypothesized that acclimation mutants should include regulatory mutants that are defective in sensing and responding to $^1O_2$. Here we describe the isolation of such a mutant and identification of a cytosolic phosphoprotein SAK1 that is critical for the acclimation and transcriptome response to $^1O_2$.

## Results

### Isolation of a singlet oxygen-sensitive mutant that is defective in acclimation

*Chlamydomonas* acclimates to singlet oxygen ($^1O_2$) generated by the exogenous photosensitizing dye rose bengal (RB) in the light (*Ledford et al., 2007*). As shown in *Figure 1A*, wild-type (WT) cells that were pretreated with RB in the light were able to survive a challenge treatment with much higher concentrations of RB, unlike cells pretreated with RB in the dark. By screening an insertional mutant population (*Dent et al., 2005*) for strains that were sensitive to $^1O_2$, we isolated a mutant called *singlet oxygen acclimation knocked-out1* (*sak1*) that is defective in acclimation to $^1O_2$ (*Figure 1A*). We have previously shown that *Chlamydomonas* WT cells can also acclimate to RB following pretreatment with high light (*Ledford et al., 2007*), indicating that high light and RB induce overlapping responses to $^1O_2$. When subjected to the same conditions (high light pretreatment followed by challenge with RB), *sak1* demonstrated less robust cross-acclimation (*Figure 1B*). We also tested conversely whether pretreatment with RB can acclimate the cells to growth in high light or in the presence of norflurazon. No increase in resistance to high light or norflurazon was induced by pretreatment with RB in either WT or *sak1* (*Figure 1—figure supplement 1*). The viability phenotypes after RB treatment shown in *Figure 1A* were paralleled by changes in $F_v/F_m$ values, a chlorophyll fluorescence parameter representing photosystem II efficiency (*Figure 1C*). In both WT and *sak1*, pretreatment did not cause an inhibition of photosystem II, as demonstrated by unchanged $F_v/F_m$ values after 30 min. However, pretreatment increased resistance of photosystem II to the RB challenge only in WT and not in *sak1* cells (*Figure 1C*). The pretreatment protected the cells only transiently, as by 90 min of challenge treatment both genotypes appeared to have experienced similar inhibition of photosystem II (*Figure 1C*), consistent with the hypothesis that *sak1* is disrupted in early sensing and/or initiation of $^1O_2$ response rather than its direct detoxification.

In contrast to its RB sensitivity, *sak1* exhibited wild-type resistance to high light, various photosynthetic inhibitors and generators of other ROS, suggesting its defect is specific to $^1O_2$ (*Figure 1D*). When tested for the gene expression response of the known $^1O_2$-specific gene *GPX5* (*Leisinger et al., 2001*) during acclimation, WT cells showed a 20- to 30-fold induction, whereas a known $H_2O_2$-responsive ascorbate peroxidase gene (*APX1*) in *Chlamydomonas* (*Urzica et al., 2012*) and a catalase gene (*CAT1*), known to be $H_2O_2$ responsive in *Arabidopsis* (*Davletova et al., 2005*; *Vanderauwera et al., 2005*), were unchanged. The mutant *sak1* showed attenuated *GPX5* induction, as expected for a mutant defective in the $^1O_2$ response (*Figure 1E*).

### The global gene expression response to $^1O_2$ in *Chlamydomonas* is distinct from that in *Arabidopsis*

To obtain insight into the cellular processes and the genes involved in $^1O_2$ acclimation, we used RNA-seq to define the transcriptome of WT cells during acclimation. The sequences were mapped to the *Chlamydomonas reinhardtii* genome version 4 (v4), and 16476 transcripts corresponding to gene models were detected (*Wakao et al., 2014*). We validated the data by quantitative reverse transcriptase PCR (qRT-PCR) for some of the differentially expressed genes during acclimation (*Figure 2*).

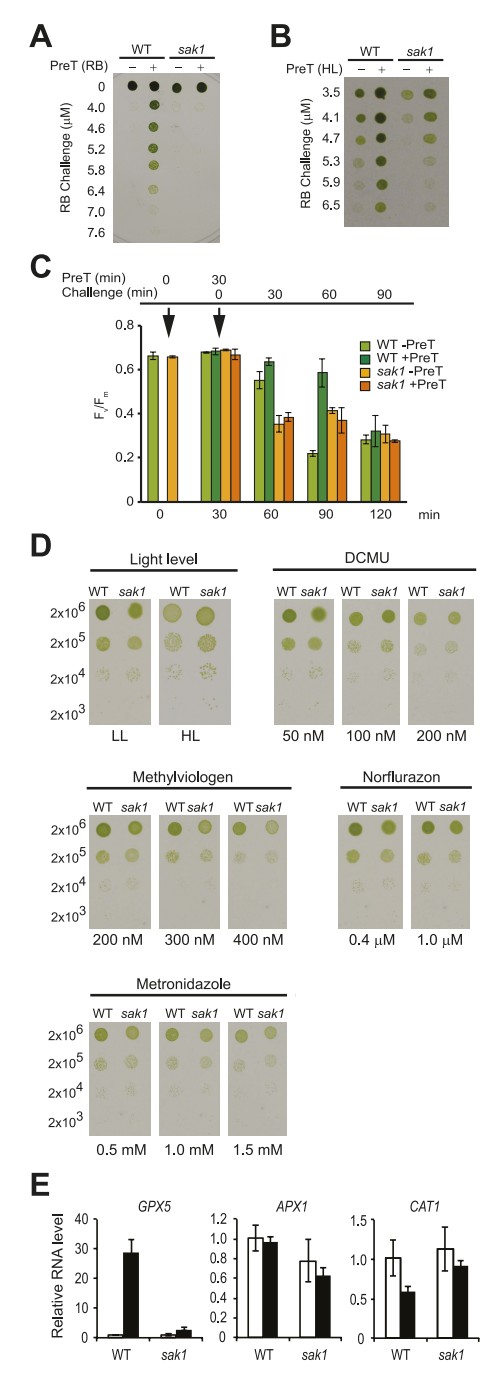

**Figure 1**. The *sak1* mutant is defective in singlet oxygen acclimation. (**A**) Acclimation phenotype of WT and *sak1*. The cells were pretreated in the dark (−) or under light (+) in the presence of rose bengal (RB), which requires light for generation of $^1O_2$. Pretreatment was followed by a subsequent higher concentration of RB (Challenge) as indicated under light. (**B**) Cells grown in low light were either kept in low light (−) or transferred to high light (+) for an hour before challenge in the light with increasing RB concentrations. (**C**) $F_v/F_m$
*Figure 1. Continued on next page*

Basal expression of some of the genes was elevated in *sak1* compared to WT (Cre16.g683400 and *GST1*, **Figure 2**). Comparisons of the fold change (FC) values obtained by RNA-seq and qRT-PCR for the genes tested in **Figure 2** are shown in **Figure 2**. The FC values are comparable between the two methods, although genes with FC greater than 20 (detected by RNA-seq) showed FC values (estimated by qRT-PCR) that were two to three times higher (Cre06.g281250.t1.1, Cre13.g566850.t1.1, Cre06.g263550.t1.1, Cre14.g623650.t1.2). Some of the genes were also induced by a transition from low light to high light, although not as strongly (**Table 1**), indicating that the $^1O_2$ response elicited by addition of RB partly overlaps with that caused by increased light intensity. To examine whether the transcriptome changes were specific to $^1O_2$, we examined the expression of several previously identified $H_2O_2$-responsive genes (**Urzica et al., 2012**) (**Table 2**). Two of the seven genes, *VTC2* (3.4-fold) and *DHAR1* (twofold) were induced during $^1O_2$ acclimation, whereas the other five genes were not differentially expressed (induced more than twofold) in our data. For these two genes, their magnitude of induction by $^1O_2$ was smaller than that of $H_2O_2$-treated cells (both genes were ~ninefold induced by 1 mM $H_2O_2$ treatment for 60 min) (**Urzica et al., 2012**). These differences suggest that our treatment with $^1O_2$ did not lead to a large-scale induction of $H_2O_2$-responsive genes, and it is likely that the two above-mentioned genes involved in ascorbate metabolism respond to both $H_2O_2$ and $^1O_2$.

During acclimation of WT to $^1O_2$, 515 genes were up-regulated at least twofold with a false discovery rate (FDR) smaller than 1% (**Supplementary file 1**, C1), and 33% of these could be categorized into functional classes based on MapMan (**Thimm et al., 2004**) using the Algal Functional Annotation Tool (**Lopez et al., 2011**) (**Figure 3A,B**). The enriched classes are marked with asterisks, and the genes within those classes are listed in **Table 3**. Genes involved in sterol/squalene/brassinosteroid metabolism (in the hormone and lipid metabolism functional classes) were notably enriched (**Table 3**). A sterol methyltransferase was also detected to display differential expression in our previous microarray analysis (**Ledford et al., 2007**). Brassinosteroids are not known to exist in *Chlamydomonas*, and in plants increasing evidence indicates sterols have a signaling role independent of brassinosteroids (**Lindsey et al., 2003**; **Boutté and Grebe, 2009**). Two cyclopropane fatty acid synthases (CFAs) were among the up-regulated lipid metabolism

*Figure 1. Continued*

values were measured after each time point indicated. Pretreatment (PreT) with 0.5 µM RB was applied for 30 min with (+PreT) or without (−PreT) light. After the pretreatment, RB was added to both dark and light samples to a final concentration of 3.75 µM RB (challenge), and $F_v/F_m$ was measured for 90 min at 30 min intervals (total 120 min). First arrow: addition of pretreatment; second arrow: addition of challenge. (**D**) *sak1* has wild-type sensitivity to other photo-oxidative stresses. Serial dilutions of WT and *sak1* were spotted onto minimal (HS) plates at the indicated light intensity or on TAP plates containing the indicated inhibitor. DCMU, 3-(3,4-dichlorophenyl)-1,1-dimethylurea; low light (LL), 80 µmol photons $m^{-2}$ $s^{-1}$; high light (HL), 450 µmol photons $m^{-2}$ $s^{-1}$. (**E**) Gene expression of a known $^1O_2$-responsive gene, *GPX5*, is induced during acclimation, while two genes associated with $H_2O_2$ response, *APX1* and *CAT1*, are not. WT cells were mock-pretreated without RB (white bars) or pretreated with RB in the light (black bars).

The following figure supplements are available for figure 1:

**Figure supplement 1**. Pretreatment with RB does not increase resistance to high light or norflurazon in cells grown on plates.

genes (*Table 3*). Another function that was notable among up-regulated genes, although they were not grouped to a common functional class by MapMan, were two genes coding for SOUL heme-binding domain proteins that were *SAK1*-dependent (SOUL2 and Cre06.g299700.t1.1, formerly annotated as SOUL1) (*Figure 2*). Genes annotated as involved in transport comprised one of the most enriched classes (*Figure 3B*). These included a number of multidrug-resistant (MDR) and pleiotropic drug-resistant (PDR) type transporters as well as other various transporters for ions, peptides, and lipids (*Table 3*). The former types of transporters may reflect the cells' response to pump RB out. When the responses to the chemical RB and $^1O_2$ were uncoupled by comparing gene expression in cultures kept in the dark with and without RB, all of the tested $^1O_2$-induced genes and ABC transporters identified from our RNA-seq remained unchanged by RB in the dark in both WT and *sak1* (*Table 4*). This result indicates that the up-regulation of these genes when RB was added in the light was a response to $^1O_2$ rather than to RB itself. Up-regulation of stress genes included those coding for chaperones and some receptor-like proteins (*Figure 3B*; *Table 3*), suggesting that the cells do mount a stress response during acclimation though not visible by gross growth phenotype (*Figure 1A*) or decrease in $F_v/F_m$ (*Figure 1C*). A smaller number of 219 genes was down-regulated during acclimation in WT (*Supplementary file 1*, C1), only 21% of which had functional annotation. The most enriched classes of down-regulated genes were nucleotide metabolism and transport, the latter including a distinct type of transporter for small metabolites and ions, different from those found among up-regulated genes that included many MDR- and PDR-type transporters (*Figure 3B*; *Table 3*).

Although only 33% of the up-regulated genes have a functional annotation (*Figure 3B*), it is interesting that the $^1O_2$ response in *Chlamydomonas* involves genes and biological processes that appear to be distinct from those that respond specifically to $^1O_2$ in *Arabidopsis* (*op den Camp et al., 2003*). A total of 70 $^1O_2$-response genes have been defined using a microarray with the *flu* mutant in *Arabidopsis* (*op den Camp et al., 2003*). These genes include the following classes (number of genes): metabolism (11), transcription (5), protein fate (4), transport (2), cellular communication/signal transduction (17), cell rescue/defense in virulence (4), subcellular localization (2), binding function or cofactor requirement (1), transport facilitation (5) and others (19). From this list of 70 genes we found four similarly annotated genes within our 515 genes induced by $^1O_2$ in *Chlamydomonas*: a Myb transcription factor, a mitochondrial carrier protein, an amino acid permease, and an ATPase/aminophospholipid translocase. None of these genes in *Chlamydomonas* was the closest ortholog of the corresponding *Arabidopsis* gene. Conversely, genes similar to those strongly up-regulated in a SAK1-dependent manner such as CFAs, SOUL proteins, GPX, and sterol biosynthetic enzymes were not found among the *Arabidopsis* $^1O_2$-specific genes despite having clear counterparts in *Arabidopsis*. Taken together, these results suggest that these two organisms may deploy distinct mechanisms in their responses to $^1O_2$.

## The *sak1* mutant is defective in the global gene expression response during acclimation to $^1O_2$

In the *sak1* mutant, 1020 genes were up-regulated, whereas 434 genes were down-regulated during acclimation (*Supplementary file 1*, C2). 350 of the 515 genes up-regulated in WT overlapped with the set of up-regulated genes in the mutant (*Figure 3A*). Comparing the fold changes of genes in WT and

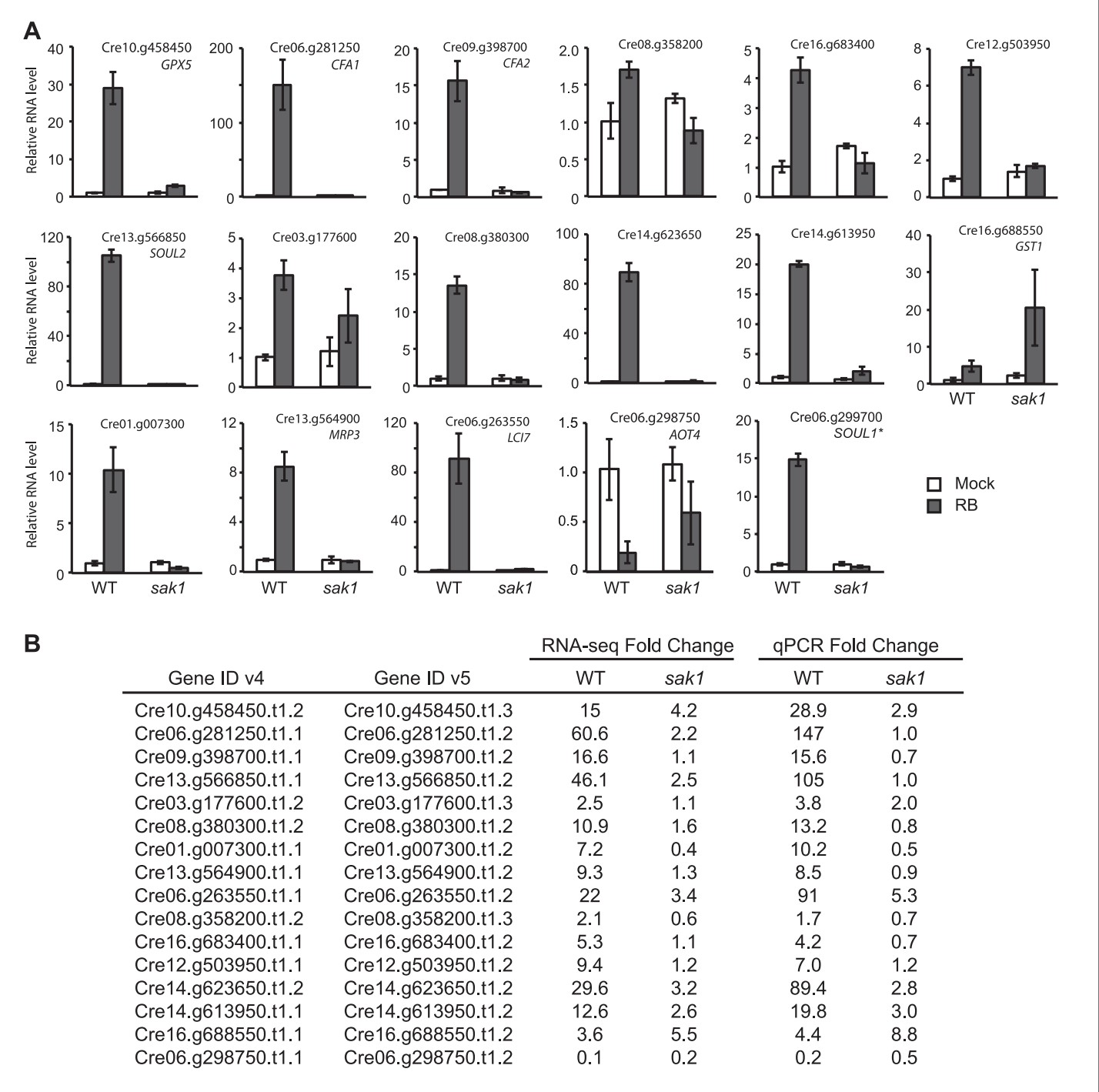

**Figure 2**. qRT-PCR analysis of genes identified to be $^1O_2$-responsive by RNA-seq. (**A**) The error bars indicate standard deviation of biological triplicates. The locus of the transcript (v5) and gene name if annotated, are indicated. *SOUL1 was named gene in v4 but not in v5. (**B**) Comparison of fold change values from RNA-seq data and qPCR. Fold change values were calculated for RNA-seq as described in 'Material and methods', and the values for qPCR are averages obtained from biological triplicates.

*sak1* during acclimation, we defined 104 genes as SAK1-dependent genes that displayed moderate to strong attenuation in their response (fold change ratio <0.5) (*Table 5*). Some of the genes that belong to enriched biological classes found among WT up-regulated genes are indicated in *Table 3*. Interestingly, the most strongly induced genes in WT were found among this group; 37 out of 104

**Table 1.** Moderate induction of $^1O_2$ genes during high light exposure

| Gene name or ID | Fold change (SD)* | |
| --- | --- | --- |
| | WT | *sak1* |
| *GPX5* | 2.86 (1.06) | 1.08 (0.23) |
| *CFA1* | 3.75 (0.99) | 1.78 (0.52) |
| *SOUL2* | 3.45 (1.25) | 1.82 (0.22) |
| *MRP3* | 3.10 (0.39) | 2.37 (0.32) |
| Cre14.g613950 | 1.42 (0.53) | 1.57 (0.46) |
| *LHCSR1*† | 14.91 (4.25) | 2.91 (1.35) |

*Fold change values are the average of biological triplicates and their standard deviations are indicated in parentheses.
†Known to have elevated expression in high light grown cells (*Peers et al., 2009*).

SAK1-dependent genes were among the top 10% most strongly induced genes (*Table 5*). 33 out of these 37 most strongly induced SAK1-dependent genes displayed strong disruption in their up-regulation; reduced to 0.01–0.25 of magnitude of fold change in *sak1* as compared to WT (*Table 5*). These results indicate SAK1 is required for the induction of the most strongly induced genes during acclimation reflecting its critical role in regulating the cellular acclimation response to $^1O_2$.

Classes of up-regulated genes in *sak1* were distinct from those of WT and included secondary metabolism of isoprenoids (*Figure 3C*; *Table 6*), precursors to photoprotective pigments such as carotenoids and tocopherols (*Li et al., 2009*). Phenylpropanoids, a group of metabolites associated with defense against stresses such as ultraviolet light and herbivores (*Maeda and Dudareva, 2012*), also represented a larger part of the response in *sak1* as compared to WT (*Figure 3C*). Another mutant-specific class of genes was cell vesicular transport, suggesting alteration in cell organization in response to the loss of *SAK1* (*Figure 3C*; *Table 6*). There were 434 genes that were down-regulated by $^1O_2$ in the *sak1* mutant (*Supplementary file 1*, C2), none of which overlapped with the set of down-regulated genes in WT, in contrast to the overlap of up-regulated genes in the two genotypes (*Figure 3A*). Enriched classes of genes included those involved in DNA, nucleotide metabolism, hormone metabolism (not of brassinosteroid) and tetrapyrrole metabolism (*Figure 3C*, *Table 6*).

To better understand the physiology of *sak1*, including the primary and secondary effects of lacking SAK1, we also focused on changes in transcript levels at the basal level, that is, without $^1O_2$ treatment. At basal level 699 genes were induced, and 737 genes were repressed in the mutant compared to WT (*Supplementary file 1*, C3), displaying the genome-wide response to the loss of *SAK1* function despite the mutant's wild-type appearance under normal lab growth conditions (*Figure 1D*). The enriched classes of genes that are differentially expressed are shown in *Figure 3D*. Genes induced in the mutant at basal level were enriched for those annotated to be involved in nucleotide metabolism, DNA, and RNA (*Figure 3D*; *Table 7*). Interestingly genes involved in tetrapyrrole and photosynthesis were enriched both in elevated and repressed genes at the basal level in *sak1*. There was no overall trend of these two pathways being up- or down-regulated, since these genes were at different steps of the pathway or encoded a select isoform of an enzyme or a subunit of a complex (*Figure 3D*; *Table 7*).

**Table 2.** Expression of $H_2O_2$ response genes during $^1O_2$ acclimation

| Gene name | Gene ID | | RPKM* | | | | Fold change† | |
| --- | --- | --- | --- | --- | --- | --- | --- | --- |
| | v4 | v5 | WT-mock | WT-RB | *sak1*-mock | *sak1*-RB | WT | *sak1* |
| *APX1* | Cre02.g087700.t1.1 | Cre02.g087700.t1.2 | 49.70 | 36.22 | 79.65 | 58.83 | 0.73 | 0.74 |
| *MSD3* | Cre16.g676150.t1.1 | Cre16.g676150.t1.2 | 0.30 | 0.18 | 0.70 | 0.17 | 0.60 | 0.25 |
| *MDAR1* | Cre17.g712100.t1.1 | Cre17.g712100.t1.2 | 35.95 | 38.30 | 33.53 | 51.34 | 1.07 | 1.53 |
| *DHAR1* | Cre10.g456750.t1.1 | Cre10.g456750.t1.2 | 20.40 | 40.93 | 25.69 | 42.18 | 2.01 | 1.64 |
| *GSH1* | Cre02.g077100.t1.1 | Cre02.g077100.t1.2 | 28.27 | 26.91 | 40.42 | 49.95 | 0.95 | 1.24 |
| *GSHR1* | Cre06.g262100.t1.2 | Cre06.g262100.t1.3 | 19.17 | 19.02 | 19.39 | 22.41 | 0.99 | 1.16 |
| *VTC2* | Cre13.g588150.t1.1 | Cre13.g588150.t1.2 | 18.16 | 62.53 | 35.10 | 103.12 | 3.44 | 2.94 |

*Average of RPKM obtained from two sequencing lanes as described in 'Material and methods'.
†Calculated as ratio of (RPKM-RB) / (RPKM-mock).

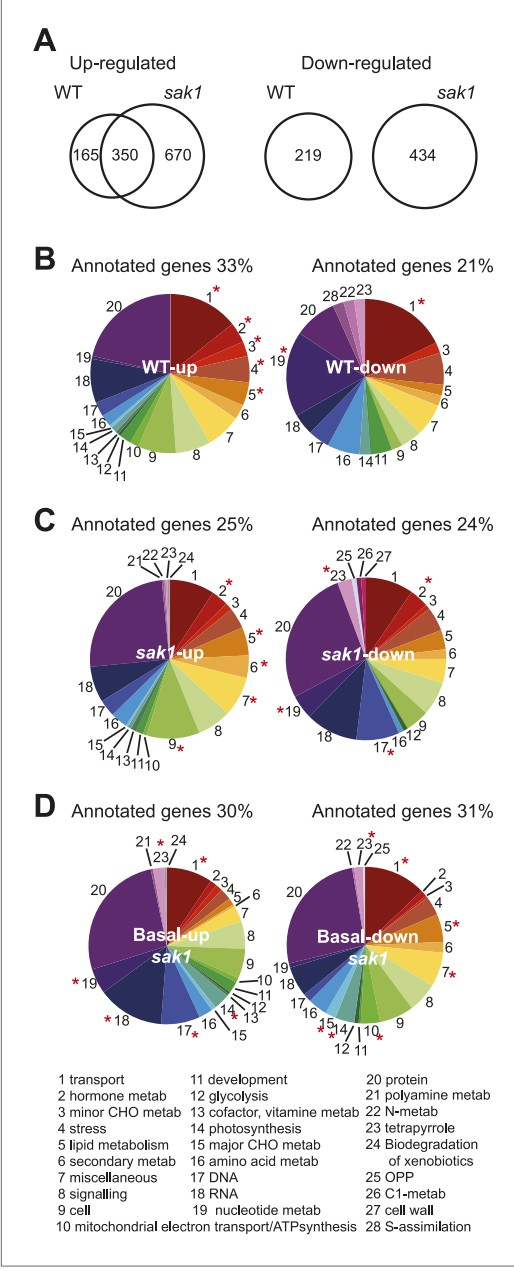

**Figure 3**. Differentially expressed genes from pair-wise comparisons. (**A**) Venn diagram representing differentially expressed genes in WT and *sak1*. Mapman functional classes distribution of differentially expressed genes (passing criteria of fold change greater than $2^1$ [up] or smaller than $2^{-1}$ [down] with FDR <1%) during acclimation in (**B**) WT and (**C**) *sak1*. (**D**) Differentially expressed genes when comparing WT and *sak1* in basal conditions (i.e., before exposure to $^1O_2$). The functional classes represented by the numbers are listed; asterisks indicate classes that were enriched compared to the genome.

We observed that some of the genes more strongly dependent on *SAK1* had repressed transcript levels (e.g., *CFA1* and *SOUL2*), indicating that *SAK1* is required for their basal expression, while others had elevated basal levels (*GPX5*), suggesting that expression of these genes is controlled also by other pathways. As is discussed in the following section, *SAK1* expression monitored by qRT-PCR followed the latter trend as the 5'UTR of the gene was elevated in the mutant (*Figure 4E*), which may be a result of response to other factors such as a possible oxidization product of $^1O_2$. The SAK1-dependent genes induced by $^1O_2$ and repressed at basal level in the mutant (i.e., those that require *SAK1* for basal expression) are indicated in *Table 5*.

## The *sak1* mutant identifies a single nuclear gene that is itself induced during acclimation to $^1O_2$

The *sak1* mutant was generated by insertional mutagenesis using a plasmid that confers resistance to zeocin (*Dent et al., 2005*). Progeny obtained from a backcross of *sak1* with WT showed that the mutation causing the RB sensitivity phenotype was linked to zeocin resistance (*Figure 4A*). The site of insertion was identified by thermal asymmetric interlaced (TAIL)-PCR (*Liu et al., 1995*) as the second exon of the annotated gene Cre17.g741300 on chromosome 17 (*Figure 4B*). To test whether this gene is responsible for the mutant phenotype, a genomic fragment containing the gene with an additional ~500 bp region upstream of the predicted transcription start site was cloned and introduced into the mutant by co-transformation. Among the approximately 300 transformants screened, two clones appeared to have recovered the RB acclimation phenotype (*Figure 4C*). Furthermore, induction of genes we found attenuated in *sak1* (*Figure 2*) was restored in these transformants (*Figure 4D*), confirming that Cre17.g741300 is the *SAK1* gene required for acclimation and the gene expression response to $^1O_2$.

In WT, the *SAK1* gene itself was induced by 6- to 10-fold during acclimation when probed for the 5'-and 3'-UTR of the transcript by qRT-PCR (*Figure 4E*). The mutant displayed elevated basal level and induction of the 5'-UTR during acclimation, whereas the 3'-UTR of the transcript was undetectable, indicating that the full-length transcript was absent in *sak1* (*Figure 4E*). An antibody raised against an epitope of the SAK1 protein detected a single band in basal conditions, whereas the SAK1 protein appeared as multiple bands with higher molecular weight in acclimated WT cells, all of which were absent in the mutant (*Figure 4F*). *SAK1* transcript was induced

**Table 3.** Enriched functional classes among differentially expressed genes in WT during $^1O_2$ acclimation

| Primary MapMan class | Secondary Mapman class | Gene ID (v4) | Gene ID (v5) | Gene name | Annotation |
|---|---|---|---|---|---|
| Up-regulated genes | | | | | |
| transport | ABC transporters and multidrug resistance systems | Cre03.g169300.t1.1 | Cre03.g169300.t2.1 | | ABC transporter (ABC-2 type) |
| | | Cre04.g220850.t1.1 | Cre04.g220850.t1.2 | | ABC transporter (ABC-2 type) |
| | | Cre11.g474600.t1.1§ | Cre02.g095151.t1 | | ABC transporter (ABC-2 type) |
| | | Cre03.g151400.t1.2 | Cre03.g151400.t1.3 | | ABC transporter (subfamilyA member3) |
| | | Cre14.g618400.t1.1§ | Cre14.g618400.t1.2 | | ABC transporter |
| | | Cre09.g395750.t1.2 | Cre09.g395750.t1.3 | | ABC transporter (plant PDR pleitropic drug resistance) |
| | | Cre14.g613950.t1.1§ | Cre14.g613950.t2.1 | | ABC transporter, Lipid exporter ABCA1 and related proteins |
| | | Cre17.g725150.t1.1 | Cre17.g725150.t1.2 | | ABC transporter |
| | | Cre04.g224400.t1.2§ | Cre04.g224400.t1.3 | | ABC transporter (plant PDR pleitropic drug resistance) |
| | | Cre13.g564900.t1.1§ | Cre13.g564900.t1.2 | MRP3 | ABC transporter, Multidrug resistance associated protein |
| | | Cre17.g721000.t1.1 | Cre17.g721000.t1.2 | | ABC transporter (ABCA) |
| | | Cre04.g224500.t1.2 | Cre04.g224500.t1.3 | | ABC transporter (plant PDR pleitropic drug resistance) |
| | | Cre01.g007000.t1.1§ | Cre01.g007000.t1.2 | | ABC transporter (ABC-2 type) |
| | unspecified anions | Cre13.g574000.t1.2 | Cre13.g574000.t1.3 | | Chloride channel 7 |
| | | Cre17.g729450.t1.1 | Cre17.g729450.t1.2 | | Chloride channel 7 |
| | amino acids | Cre04.g226150.t1.2 | Cre04.g226150.t1.3 | AOC1 | Amino acid carrier 1; belongs to APC (amino acid polyamine organocation) family |
| | misc | Cre16.g683400.t1.1§ | Cre16.g683400.t1.2 | | CRAL/TRIO domain (Retinaldehyde binding protein-related) |
| | | Cre17.g718100.t1.1 | Cre17.g718100.t1.2 | | Phosphatidylinositol transfer protein SEC14 and related proteins (CRAL/TRIO) |
| | | Cre06.g311000.t1.2 | Cre06.g311000.t1.3 | FBT2 | Folate transporte |
| | calcium | Cre09.g410050.t1.1§ | Cre09.g410050.t1.2 | | Ca2+ transporting ATPase |
| | potassium | Cre07.g329882.t1.2 | Cre07.g329882.t1.3 | | Ca2+-activated K+ channel proteins |
| | phosphate | Cre16.g686750.t1.1 | Cre16.g686750.t1.2 | PTA3 | Proton/phosphate symporter |
| | metal | Cre13.g570600.t1.1 | Cre13.g570600.t1.2 | CTR1 | CTR type copper ion transporter |
| | metabolite transporters at the mitochondrial membrane | Cre06.g267800.t1.2 | Cre06.g267800.t2.1 | | Mitochondrial carrier protein |
| hormone metabolism* | brassinosteroid | Cre16.g663950.t1.1 | Cre16.g663950.t1.2 | | Sterol C5-desaturase |

*Table 3. Continued on next page*

*Table 3. Continued*

| Primary MapMan class | Secondary Mapman class | Gene ID (v4) | Gene ID (v5) | Gene name | Annotation |
|---|---|---|---|---|---|
| | | Cre02.g076800.t1.1 | Cre02.g076800.t1.2 | | delta14-sterol reductase |
| | | Cre12.g557900.t1.1 | Cre12.g557900.t1.1 | CDI1 | C-8,7 sterol isomerase |
| | | Cre02.g092350.t1.1 | Cre02.g092350.t1.2 | | Cytochrome P450, CYP51 Sterol-demethylase |
| | | Cre12.g500500.t1.2 | Cre12.g500500.t2.1 | | SAM-dependent methyltransferases |
| | jasmonate | Cre19.g756100.t1.1 | Cre03.g210513.t1 | | 12-oxophytodienoic acid reductase |
| | auxin | Cre14.g609900.t1.1 | Cre14.g609900.t1.1 | | Predicted membrane protein, contains DoH and Cytochrome b-561/ferric reductase transmembrane domains |
| | | Cre06.g276050.t1.1 | Cre06.g276050.t1.2 | | Aldo/keto reductase |
| | | Cre16.g692800.t1.2 | Cre16.g692800.t1.3 | | Aldo/keto reductase |
| | | Cre03.g185850.t1.2 | Cre03.g185850.t1.2 | | pfkB family, sugar kinase-related |
| minor CHO metabolism | others | Cre06.g276050.t1.1 | Cre06.g276050.t1.2 | | Aldo/keto reductase |
| | | Cre16.g692800.t1.2 | Cre16.g692800.t1.3 | | Aldo/keto reductase |
| | | Cre03.g185850.t1.2 | Cre03.g185850.t1.2 | | pfkB family, sugar kinase-related |
| | callose | Cre06.g302050.t1.1 | Cre06.g302050.t1.2 | | 1,3-beta-glucan synthase |
| | myo-inositol | Cre03.g180250.t1.1 | Cre03.g180250.t1.2 | | Myo-inositol-1-phosphate synthase |
| stress | biotic | Cre01.g057050.t1.1§ | Cre03.g144324.t1 | | Leucine Rich Repeat |
| | | Cre01.g016200.t1.2 | Cre01.g016200.t1 | | Mlo Family |
| | | Cre28.g776450.t1.1§ | Cre08.g358573.t1 | PSMD10 | 26S proteasome regulatory complex |
| | abiotic | Cre12.g501500.t1.1 | NF† | | |
| | | Cre02.g132300.t1.2 | Cre09.g395732.t1 | | DnaJ domain |
| | | Cre07.g339650.t1.2 | Cre07.g339650.t1.3 | DNJ20 | DnaJ-like protein |
| | | Cre01.g033300.t1.1§ | Cre01.g033300.t2.1 | | No annotation‡ |
| | | Cre16.g677000.t1.1 | Cre16.g677000.t1.2 | HSP70E | Heat shock protein 70E |
| | | Cre08.g372100.t1.1 | Cre08.g372100.t1.2 | HSP70A | Heat shock protein 70A |
| lipid metabolism | phospholipid synthesis | Cre13.g604700.t1.2 | Cre13.g604700.t1.3 | PCT1 | CDP-alcohol phosphatidyltransferase/ Phosphatidylglycerol-phosphate synthase |
| | | Cre06.g281250.t1.1§ | Cre06.g281250.t1.2 | CFA1 | Cyclopropane fatty acid synthase |
| | | Cre09.g398700.t1.1§ | Cre09.g398700.t1.2 | CFA2 | Cyclopropane fatty acid synthase |
| | 'exotics' (steroids, squalene etc) | Cre01.g061750.t1.1 | Cre03.g146507.t1 | SPT2 | Serine palmitoyltransferase |
| | | Cre83.g796250.t1.1 | NF† | SPT1 | Serine palmitoyltransferase |
| | | Cre02.g137850.t1.1 | Cre09.g400516.t1 | | TRAM (translocating chain-associating membrane) superfamily |
| | FA synthesis and FA elongation | Cre03.g182050.t1.1 | Cre03.g182050.t1 | | Long-chain acyl-CoA synthetases (AMP-forming) |

*Table 3. Continued on next page*

*Table 3. Continued*

| Primary MapMan class | Secondary Mapman class | Gene ID (v4) | Gene ID (v5) | Gene name | Annotation |
|---|---|---|---|---|---|
| | | Cre06.g256750.t1.1 | Cre06.g256750.t1.2 | | Acyl-ACP thioesterase |
| misc | short chain dehydrogenase/reductase (SDR) | Cre12.g556750.t1.2 | Cre12.g556750.t1.3 | | Short chain dehydrogenase |
| | | Cre27.g775000.t1.1 | Cre12.g549852.t1 | | Short chain dehydrogenase |
| | | Cre17.g731350.t1.2 | Cre17.g731350.t1.2 | | Short chain dehydrogenase |
| | | Cre08.g381510.t1.1§ | NF† | | Short chain alcohol dehydrogenase |
| | UDP glucosyl and glucoronyl transferases | Cre02.g144050.t1.1 | Cre02.g144050.t2.1 | | Acetylglucosaminyltransferase EXT1/exostosin 1 |
| | | Cre16.g659450.t1.1 | Cre16.g659450.t1.2 | | Lactosylceramide 4-alpha-Galactosyltransferase |
| | | Cre03.g173300.t1.1 | Cre03.g173300.t1.2 | | Lactosylceramide 4-alpha-Galactosyltransferase |
| | dynamin | Cre02.g079550.t1.1 | Cre02.g079550.t1.2 | | Dynamin-related GTPase, involved in circadian rhythms |
| | misc2 | Cre06.g258600.t1.1§ | Cre06.g258600.t2.1 | | Predicted hydrolase related to dienelactone hydrolase |
| | acid and other phosphatases | Cre06.g249800.t1.1 | Cre06.g249800.t1.2 | | Sphingomyelin synthase |
| Down-regulated genes | | | | | |
| nucleotide metabolism | salvage | Cre13.g573800.t1.1 | Cre13.g573800.t1.2 | | Phosphoribulokinase / Uridine kinase family |
| | synthesis | Cre12.g503300.t1.1 | Cre12.g503300.t1.2 | | Phosphoribosylamidoimidazole-succinocarboxamide synthase |
| | | Cre06.g308500.t1.1 | Cre06.g308500.t1.2 | CMP2 | Carbamoyl phosphate synthase, small subunit |
| | | Cre14.g614300.t1.1 | Cre14.g614300.t1.2 | | Inosine-5-monophosphate dehydrogenase |
| transport | ABC transporters and multidrug resistance systems | Cre06.g273750.t1.2 | Cre06.g273750.t1.3 | SUA1 | Chloroplast sulfate transporter |
| | | Cre02.g083354.t1.1 | Cre02.g083354.t1 | | ATP-binding cassette, subfamily B (MDR/TAP), member 9 |
| | calcium | Cre06.g263950.t1.2 | Cre06.g263950.t1.3 | | Na+/K + ATPase, alpha subunit |
| | metabolite transporters at the envelope membrane | Cre08.g363600.t1.1 | Cre08.g363600.t1.2 | | Glucose-6-phosphate, PEP/phosphate antiporter |
| | metal | Cre17.g720400.t1.2 | Cre17.g720400.t1.3 | HMA1 | Heavy metal transporting ATPase |
| | P- and V-ATPases | Cre10.g459200.t1.1 | Cre10.g459200.t1.2 | ACA4 | Plasma membrane H + -transporting ATPase |
| | phosphate | Cre02.g144650.t1.1 | Cre02.g144650.t1.2 | PTB12 | Na+/Pi symporter |
| | potassium | Cre06.g278700.t1.2 | Cre06.g278700.t1.2 | | Myotrophin and similar proteins |

*Functional terms are inferred by homology to the annotation set of *Arabidopsis thaliana* (***Lopez et al., 2011***).
†Corresponding gene model was not found in v5.
‡No functional annotations found on v5 but defined by MapMan on Algal Functional Annotation Tool (***Lopez et al., 2011***).
§Induction during $^1O_2$ acclimation dependent on SAK1 (***Table 5***).

**Table 4.** $^1O_2$ response genes are not induced when RB is added in the dark

| Gene name or ID | Fold change +RB/−RB (SD)* | |
|---|---|---|
| | **WT** | *sak1* |
| *GPX5* | 1.13 (0.33) | 0.87 (0.31) |
| *SAK1* | 1.38 (0.08) | 1.29 (0.19) |
| *CFA1* | 0.90 (0.04) | 1.44 (0.22) |
| *SOUL2* | 1.17 (0.25) | 1.11 (0.19) |
| *MRP3*†,‡ | 1.13 (0.12) | 1.07 (0.25) |
| Cre12.g503950†,‡ | 0.93 (0.06) | 1.20 (0.12) |
| Cre14.g613950†,§ | 0.65 (0.06) | 0.79 (0.15) |
| Cre04.g220850†,‡ | 1.00 (0.09) | 1.29 (0.04) |
| Cre09.g395750†,‡ | 1.05 (0.10) | 1.29 (0.12) |

*Average of fold change and standard deviation (SD) of biological triplicates.
†Annotated as transport function.
‡ABC transporter.
§Sec14-like phosphatidylinositol transfer protein.

when probed for the 5'-UTR during high light exposure in both WT and *sak1* (*Figure 4G*) similarly to other $^1O_2$-response genes identified by RNA-seq (*Table 1*), indicating that *SAK1* itself is part of the endogenous response to high light.

## SAK1 contains an uncharacterized domain conserved in chlorophytes and found in some bZIP transcription factors

The predicted SAK1 protein consists of 1141 amino acid residues and has no domains with functional annotation. Only a ~150-residue region at the C-terminus, designated the SAK1 domain, has similarity to other proteins. Many predicted proteins within chlorophytes (*Volvox carteri* [8 proteins], *Coccomyxa subellipsoidea* [3 proteins], *Chlamydomonas* [14 proteins], *Chlorella variabilis* [9 proteins] and *Micromonas* [3 proteins]) (*Table 8*) contain this domain as shown in the alignment in *Figure 5—figure supplement 1*. Among the 37 members of the chlorophyte SAK1 domain family, 13 have possible bZIP transcription factor domains (six were significant Pfam hits and seven were below the threshold for significance but recognizable by Pfam) (*Figure 5*). One protein contained a mitochondrial (transcription) termination factor (mTERF) domain (*Figure 5*), defined by its three leucine zipper domains required for DNA binding (*Fernandez-Silva et al., 1997*). Proteins with more distantly related SAK1 domains were found by PSI-BLAST in plants, many of which were hypothetical or unknown proteins but also included bZIP transcription factors.

Amino acid positions 900 to 1089 of SAK1, corresponding to the region aligned with other proteins in *Figure 5—figure supplement 1*, were searched for secondary structure using PHYRE, and this region was predicted to consist of mostly alpha helices with some disordered intervals. The top hit was a cobalt/nickel-binding resistance protein cnrr, and 44% of the residues were modeled with 73.6% confidence (*Figure 5—figure supplement 2*).

## SAK1 resides mainly in the cytosol and is phosphorylated during induction by $^1O_2$

To obtain insight into the function of SAK1, we isolated subcellular fractions enriched for chloroplast, ER, cytosol, and mitochondria from WT cells. The *Chlamydomonas* cell contains a single large chloroplast that is physically connected to other organelles such as the ER, making it particularly challenging to fractionate. The patterns of markers specific for chloroplast, ER, cytosol, and mitochondria showed that each target fraction was enriched as expected, although with some cross contamination (*Figure 6A,B*). The distribution of SAK1 in these fractions resembled most closely that of the cytosolic marker NAB1 (*Mussgnug et al., 2005*), although the SAK1 signal was not as enriched as NAB1 in the cytosolic fraction, possibly due to partial degradation of SAK1 during the fractionation. The localization was the same in cells with and without RB treatment (*Figure 6A*). Because *SAK1* was required for the induction of many genes during acclimation to $^1O_2$ and the list of proteins with similarity to SAK1 included those predicted to be bZIP transcription factors, we tested whether SAK1 protein was dually targeted to the nucleus and cytosol, which would account for the lack of enrichment of SAK1 in the cytosolic fraction (*Figure 6A*). As shown in *Figure 6C* although a faint SAK1 signal was detected in nuclear fraction, there was no enrichment as seen for the nuclear marker histone H3 (H3). The distribution of the cytosolic marker NAB1 indicated the contamination of the nuclear fraction by cytosolic proteins (*Figure 6C*). Therefore we conclude that the low signal of SAK1 in the nuclear fraction is likely to be due to cytosolic contamination. Attempts to detect the protein by immunofluorescence using anti-SAK1 antibodies as well as anti-FLAG and anti-HA antibodies against tagged proteins in transgenic lines were unsuccessful due to a very low signal-to-noise ratio even in bleached cells.

**Table 5.** Genes that require SAK1 for induction by $^1O_2$

| Gene ID (v4) | Gene ID (v5) | Gene name | Annotation | FC WT* (log₂) | FC sak1 (log₂) | Attenuation (FC-sak1/FC-WT)† | Basal repression in sak1 (log₂) |
|---|---|---|---|---|---|---|---|
| Cre02.g137700.t1.1‡ | Cre09.g400404 | | | 6.49 | 1.80 | 0.04 | −3.35 |
| Cre06.g281250.t1.1‡ | Cre06.g281250 | CFA1 | Cyclopropane fatty acid synthase | 5.92 | 1.16 | 0.04 | −2.10 |
| Cre27.g775950.t1.2 | Cre12.g557928 | | | 5.83 | 0.81 | 0.03 | |
| Cre01.g033300.t1.1 | Cre01.g033300 | | | 5.72 | −0.39 | 0.01 | |
| Cre13.g566850.t1.1‡ | Cre13.g566850 | SOUL2 | SOUL heme-binding protein | 5.53 | 1.33 | 0.05 | −2.60 |
| Cre14.g623650.t1.1 | Cre14.g623650 | | Alcohol dehydrogenase | 4.89 | 1.67 | 0.11 | |
| Cre13.g600650.t1.1 | Cre06.g278245 | | Rieske 2Fe-2S domain | 4.76 | 1.64 | 0.12 | |
| Cre06.g263550.t1.1 | Cre06.g263550 | LCI7 | R53.5-related protein | 4.46 | 1.77 | 0.15 | |
| Cre07.g342100.t1.1 | Cre07.g342100 | | | 4.43 | 1.40 | 0.12 | |
| Cre06.g299700.t1.1‡ | Cre06.g299700 | SOUL1 | SOUL heme-binding protein | 4.32 | 0.43 | 0.07 | −1.13 |
| Cre09.g398700.t1.1‡ | Cre09.g398700 | CFA2 | Cyclopropane fatty acid synthase | 4.05 | 0.18 | 0.07 | −1.00 |
| Cre12.g492650.t1.1‡ | Cre12.g492650 | FAS2 | Fasciclin-like protein | 4.01 | 0.07 | 0.07 | −1.24 |
| Cre08.g381510.t1.1 | NF | | | 3.94 | 0.73 | 0.11 | |
| Cre10.g458450.t1.2 | Cre10.g458450 | GPX5 | Glutathione peroxidase | 3.91 | 2.06 | 0.28 | |
| Cre11.g474600.t1.1 | Cre02.g095151 | | ABC transporter (ABC-2 type) | 3.90 | 0.44 | 0.09 | |
| Cre13.g600700.t1.1 | Cre06.g278246 | | | 3.78 | 1.48 | 0.20 | |
| Cre14.g613950.t1.1 | Cre14.g613950 | | | 3.65 | 1.38 | 0.21 | |
| Cre06.g269300.t1.1 | Cre06.g269300 | | DUF1365 | 3.50 | 0.40 | 0.12 | |
| Cre08.g380300.t1.2 | Cre08.g380300 | MSRA3 | Peptide methionine sulfoxide reductase | 3.45 | 0.66 | 0.14 | |
| Cre28.g776450.t1.1 | Cre08.g358573 | TRP7 | Transient receptor potential ion channel | 3.31 | −0.79 | 0.06 | |
| Cre01.g031650.t1.2 | Cre01.g031650 | CGLD12 | Potential galactosyl transferase activity | 3.30 | 0.67 | 0.16 | |
| Cre14.g629061.t1.1 | NF | | DUF2177 | 3.25 | 0.08 | 0.11 | |
| Cre12.g503950.t1.1 | Cre12.g503950 | | CRAL/TRIO domain | 3.24 | 0.31 | 0.13 | |

*Table 5. Continued on next page*

*Table 5. Continued*

| Gene ID (v4) | Gene ID (v5) | Gene name | Annotation | FC WT* ($\log_2$) | FC *sak1* ($\log_2$) | Attenuation (FC-*sak1*/FC-WT)† | Basal repression in *sak1* ($\log_2$) |
|---|---|---|---|---|---|---|---|
| Cre13.g564900.t1.1 | Cre13.g564900 | | ABC transporter transmembrane region | 3.22 | 0.34 | 0.14 | |
| Cre02.g139500.t1.1 | Cre09.g401701 | | DUF1295 | 3.04 | −0.16 | 0.11 | |
| Cre14.g618400.t1.1 | Cre14.g618400 | | | 2.97 | 1.15 | 0.28 | |
| Cre17.g715150.t1.1 | Cre17.g715150 | | | 2.89 | 0.13 | 0.15 | |
| Cre17.g741300.t1.2‡ | Cre17.g741300 | SAK1 | | 2.88 | 0.66 | 0.21 | −2.77 |
| Cre01.g007300.t1.1 | Cre01.g007300 | | | 2.85 | −1.15 | 0.06 | |
| Cre16.g648700.t1.2‡ | Cre16.g648700 | | ABC transporter (ABC-2 type) | 2.79 | 0.26 | 0.17 | −1.26 |
| Cre13.g566900.t1.2 | Cre13.g566900 | | | 2.76 | −0.38 | 0.11 | |
| Cre02.g137750.t1.2 | Cre09.g400441 | | JmjC domain | 2.72 | −0.31 | 0.12 | |
| Cre06.g263500.t1.1 | Cre06.g263500 | | Archease protein family (DUF101) | 2.67 | 1.02 | 0.32 | |
| Cre01.g016150.t1.1‡ | Cre01.g016150 | | ADP-ribosylglycohydrolase | 2.65 | 0.17 | 0.18 | −1.26 |
| Cre08.g380000.t1.1 | Cre08.g380000 | | Formylglycine-generating sulfatase enzyme | 2.59 | 1.53 | 0.48 | |
| Cre14.g615600.t1.1 | Cre14.g615600 | | Putative serine esterase (DUF676) | 2.53 | −0.54 | 0.12 | |
| Cre11.g472900.t1.2 | Cre02.g095113 | | CAP-Gly domain | 2.45 | −0.05 | 0.18 | |
| Cre06.g269250.t1.1 | Cre06.g269250 | | | 2.44 | 0.55 | 0.27 | |
| Cre02.g120600.t1.1 | Cre09.g403071 | | | 2.44 | 0.94 | 0.35 | |
| Cre06.g261200.t1.1 | Cre06.g261200 | ERG25 | Sterol desaturase | 2.42 | 0.64 | 0.29 | |
| Cre16.g683400.t1.1 | Cre16.g683400 | | CRAL/TRIO domain | 2.40 | 0.08 | 0.20 | |
| Cre22.g765150.t1.1 | Cre11.g467725 | | hypothetical protein | 2.30 | 0.46 | 0.28 | |
| Cre13.g571800.t1.2 | Cre13.g571800 | | DUF1336 | 2.27 | 0.72 | 0.34 | |
| Cre13.g579450.t1.2 | Cre13.g579450 | CST1 | Membrane transporter | 2.27 | 1.23 | 0.49 | |
| Cre08.g380350.t1.1 | Cre08.g380350 | | | 2.21 | −0.01 | 0.21 | |
| Cre16.g649250.t1.2 | Cre16.g649250 | | | 2.08 | 0.58 | 0.35 | |

Table 5. Continued

| Gene ID (v4) | Gene ID (v5) | Gene name | Annotation | FC WT* (log₂) | FC sak1 (log₂) | Attenuation (FC-sak1/FC-WT)† | Basal repression in sak1 (log₂) |
|---|---|---|---|---|---|---|---|
| Cre11.g476250.t1.1 | Cre11.g476250 | | | 2.08 | 0.49 | 0.33 | |
| Cre02.g108000.t1.2 | Cre02.g108000 | | | 2.08 | 1.03 | 0.49 | |
| Cre13.g583300.t1.1 | Cre13.g583300 | | | 1.98 | −0.48 | 0.18 | |
| Cre04.g215300.t1.2 | NF | | | 1.97 | 0.57 | 0.38 | |
| Cre02.g139450.t1.1 | Cre02.g139450 | | DUF947 | 1.95 | −0.62 | 0.17 | |
| Cre03.g194750.t1.2 | Cre03.g194750 | | | 1.95 | 0.73 | 0.43 | |
| Cre06.g258600.t1.1 | Cre06.g258600 | | Dienelactone hydrolase family | 1.91 | −0.95 | 0.14 | |
| Cre10.g418700.t1.1 | Cre10.g418700 | | Probable N6-adenine methyltransferase | 1.87 | −0.03 | 0.27 | |
| Cre10.g444550.t1.1 | Cre10.g444550 | SPP1A | Signal peptide peptidase | 1.81 | 0.51 | 0.41 | |
| Cre01.g060050.t1.2 | Cre03.g145807 | | | 1.78 | −0.11 | 0.27 | |
| Cre09.g410050.t1.1 | Cre09.g410050 | | Calcium transporting ATPase | 1.76 | 0.51 | 0.42 | |
| Cre03.g163400.t1.2 | Cre03.g163400 | | | 1.76 | −0.17 | 0.26 | |
| Cre01.g008450.t1.1 | Cre01.g008450 | | Nuf2 family | 1.73 | −0.54 | 0.21 | |
| Cre12.g536650.t1.1 | Cre12.g536650 | | | 1.72 | 0.35 | 0.39 | |
| Cre02.g114900.t1.2 | Cre02.g114900 | ANK23 | predicted protein | 1.71 | 0.08 | 0.32 | |
| Cre16.g661850.t1.2 | Cre16.g661850 | | Calcium/calmoduline dependent protein kinase association | 1.69 | 0.03 | 0.32 | |
| Cre14.g615500.t1.2 | Cre14.g615500 | | Glycoprotease family | 1.68 | −0.76 | 0.18 | |
| Cre11.g483100.t1.2 | Cre11.g483100 | | Protein kinase | 1.66 | −0.49 | 0.22 | |
| Cre28.g776650.t1.1 | Cre08.g358569 | | | 1.64 | 0.33 | 0.40 | |
| Cre07.g340250.t1.2 | Cre07.g340250 | | Protein kinase | 1.63 | −0.41 | 0.24 | |
| Cre06.g296250.t1.2 | Cre06.g296250 | SYK1 | tRNA synthetase, class II | 1.60 | 0.54 | 0.48 | |
| Cre06.g310500.t1.1 | Cre06.g310500 | | | 1.57 | 0.18 | 0.38 | |
| Cre07.g342800.t1.2 | Cre07.g342800 | CGL16 | Predicted protein | 1.49 | 0.32 | 0.44 | |

Table 5. Continued on next page

Table 5. Continued

| Gene ID (v4) | Gene ID (v5) | Gene name | Annotation | FC WT* (log$_2$) | FC sak1 (log$_2$) | Attenuation (FC-sak1/FC-WT)† | Basal repression in sak1 (log$_2$) |
|---|---|---|---|---|---|---|---|
| Cre03.g181450.t1.2 | Cre03.g181450 | | DUF1619 | 1.47 | 0.35 | 0.46 | |
| Cre66.g793601.t1.1 | Cre35.g759497 | | | 1.47 | 0.03 | 0.37 | |
| Cre14.g614050.t1.2 | Cre14.g614050 | MAP65 | Microtubule associated protein | 1.43 | 0.06 | 0.39 | |
| Cre04.g217500.t1.1 | Cre04.g217500 | | Inosine-uridine preferring nucleoside hydrolase | 1.42 | 0.19 | 0.43 | |
| Cre06.g292950.t1.1 | Cre06.g292950 | | DNA polymerase delta, subunit 4 | 1.38 | −0.12 | 0.35 | |
| Cre16.g661750.t1.1 | Cre16.g661750 | | Calcium/calmoduline dependent protein kinase association | 1.38 | −0.12 | 0.35 | |
| Cre01.g007000.t1.1 | Cre01.g007000 | | ABC transporter (ABC-2 type) | 1.35 | 0.21 | 0.45 | |
| Cre04.g224400.t1.2 | Cre04.g224400 | | ABC transporter (ABC-2 type) | 1.34 | −0.13 | 0.36 | |
| Cre01.g068400.t1.2 | Cre16.g680790 | | | 1.33 | 0.16 | 0.45 | |
| Cre05.g237400.t1.1 | Cre05.g237400 | DAE1 | Diaminopimelate epimerase | 1.32 | 0.22 | 0.47 | |
| Cre14.g609600.t1.2 | Cre14.g609600 | | | 1.32 | −0.58 | 0.27 | |
| Cre05.g234850.t1.2 | Cre05.g234850 | | Ubiquitin carboxyl-terminal hydrolase | 1.29 | 0.16 | 0.46 | |
| Cre03.g179200.t1.1 | Cre03.g179200 | | | 1.28 | −0.48 | 0.30 | |
| Cre10.g417730.t1.1 | Cre10.g417730 | | | 1.27 | 0.17 | 0.47 | |
| Cre03.g159700.t1.2 | Cre03.g159700 | | | 1.26 | −0.14 | 0.38 | |
| Cre12.g540150.t1.2 | Cre12.g540150 | | | 1.19 | −0.24 | 0.37 | |
| Cre01.g006550.t1.2‡ | Cre01.g006550 | | No annotation | 1.17 | −0.49 | 0.32 | −1.60 |
| Cre03.g159950.t1.2 | Cre03.g159950 | | | 1.17 | −0.17 | 0.40 | |
| Cre27.g775900.t1.2 | Cre12.g557503 | | | 1.14 | −0.70 | 0.28 | |
| Cre02.g121600.t1.1 | Cre09.g387208 | | Protein kinase | 1.14 | 0.00 | 0.46 | |
| Cre14.g609550.t1.1 | NF | | | 1.13 | −0.84 | 0.26 | |
| Cre07.g315050.t1.2 | Cre07.g315050 | | | 1.12 | −0.03 | 0.45 | |
| Cre04.g218800.t1.2 | Cre04.g218800 | THB3 | Truncated hemoglobin | 1.11 | −0.50 | 0.33 | |

Table 5. Continued on next page

Table 5. Continued

| Gene ID (v4) | Gene ID (v5) | Gene name | Annotation | FC WT* (log$_2$) | FC sak1 (log$_2$) | Attenuation (FC-sak1/FC-WT)† | Basal repression in sak1 (log$_2$) |
|---|---|---|---|---|---|---|---|
| Cre02.g133300.t1.1 | Cre09.g396624 | | | 1.11 | −0.43 | 0.34 | |
| Cre01.g060650.t1.2 | Cre03.g146067 | | | 1.10 | −0.42 | 0.35 | |
| Cre01.g057050.t1.1 | Cre03.g144324 | | | 1.10 | 0.04 | 0.48 | |
| Cre06.g304950.t1.1 | Cre06.g304950 | | | 1.07 | −0.65 | 0.30 | |
| Cre08.g358200.t1.2 | Cre08.g358200 | A4 | Protein kinase | 1.07 | −0.82 | 0.27 | |
| Cre16.g689550.t1.2 | Cre16.g689550 | PTK8 | Putative tyrosine kinase | 1.06 | −0.17 | 0.43 | |
| Cre17.g720950.t1.1 | Cre17.g720950 | | 3-oxo-5-alpha-steroid 4-dehydrogenase | 1.05 | −0.26 | 0.40 | |
| Cre02.g090950.t1.2 | Cre02.g090950 | | | 1.05 | −0.27 | 0.40 | |
| Cre16.g683350.t1.1 | Cre16.g683350 | | | 1.03 | −0.67 | 0.31 | |
| Cre02.g109450.t1.1 | Cre02.g109450 | | | 1.01 | −0.03 | 0.48 | |
| Cre16.g652750.t1.1 | Cre16.g652750 | | | 1.01 | −0.29 | 0.41 | |
| Cre03.g190000.t1.1 | Cre03.g190000 | | | 1.00 | −0.99 | 0.25 | |

*Data were ordered by FC in WT.

†Of the 52 most highly induced genes in WT (the top 10%), 37 were SAK1-dependent, and the induction of 33 of these genes was strongly attenuated to only 0.01-0.25 of magnitude of FC found in the WT. Dashed line indicates cutoff of FC for the top 10% most strongly induced genes.

‡Genes that are repressed at basal level in sak1.

NF, not found in v5.

**Table 6.** Enriched functional classes among differentially expressed genes in *sak1* during $^1O_2$ acclimation

| Primary Mapman class | Secondary Mapman class | Gene ID (v4) | Gene name | Annotation |
|---|---|---|---|---|
| Up-regulated genes | | | | |
| Secondary metabolism | isoprenoids | Cre13.g565650.t1.1 | | Geranylgeranyl pyrophosphate synthase/Polyprenyl synthetase |
| | | Cre06.g267600.t1.1 | | Lycopene epsilon cyclase |
| | | Cre09.g407200.t1.1 | | Phytoene desaturase |
| | | Cre06.g267600.t1.1 | | Lycopene epsilon cyclase |
| | | Cre01.g011100.t1.1 | | Prenyltransferase and squalene oxidase repeat, Oxidosqualene-lanosterol cyclase and related proteins |
| | N misc | Cre08.g381707.t1.1 | | NF* |
| | phenylpropanoids | Cre03.g207800.t1.1 | | Alcohol dehydrogenase, class V |
| | | Cre14.g623650.t1.1 | | Alcohol dehydrogenase, class V (Zinc-binding) |
| | | Cre01.g039350.t1.1 | | Cytochrome P450 reductase, possibly CYP505B family |
| | sulfur-containing | Cre06.g299400.t1.1 | | NF* |
| | wax | Cre17.g722150.t1.1 | *PKS3* | Type III polyketide synthase |
| | | Cre07.g318500.t1.2 | | FAE1/Type III polyketide synthase-like protein, Chalcone and stilbene synthases |
| Lipid metabolism | 'exotics' (steroids, squalene etc) | Cre01.g061750.t1.1 | | serine palmitoyltransferase |
| | | Cre02.g137850.t1.1 | | NF* |
| | | Cre83.g796250.t1.1 | | NF* |
| | | Cre01.g011100.t1.1 | | Prenyltransferase and squalene oxidase repeat, Oxidosqualene-lanosterol cyclase and related proteins |
| | FA synthesis and FA elongation | Cre06.g256750.t1.1 | | Acyl carrier protein thioesterase |
| | | Cre03.g182050.t1.1 | | Long-chain acyl-CoA synthetases (AMP-forming) |
| | | Cre02.g074650.t1.1 | | Kelch repeat-containing proteins, Acyl-CoA binding protei |
| | glycerol metabolism | Cre01.g053000.t1.1 | *GPD2* | Glycerol-3-phosphate dehydrogenase/dihydroxyacetone-3-phosphate reductase |
| | glycolipid synthesis | Cre13.g583600.t1.1 | *DGD1* | Digalactosyldiacylglycerol synthase |
| | lipid degradation | Cre01.g057450.t1.2 | | NF* |
| | | Cre02.g126050.t1.1 | | NF* |
| | phospholipid synthesis | Cre06.g281250.t1.1 | *CFA1* | Cyclopropane fatty acid synthase |
| | | Cre01.g038250.t1.1 | *SDC1* | Serine decarboxylase |
| | | Cre11.g472700.t1.1 | | NF* |
| | | Cre13.g604700.t1.2 | | CDP-alcohol phosphatidyltransferase/ Phosphatidylglycerol-phosphate synthase |

*Table 6. Continued on next page*

Table 6. Continued

| Primary Mapman class | Secondary Mapman class | Gene ID (v4) | Gene name | Annotation |
|---|---|---|---|---|
| Cell | vesicle transport | Cre18.g744100.t1.1 | | NF* |
| | | Cre17.g721900.t1.1 | COG5 | Component of oligomeric golgi complex |
| | | Cre01.g003050.t1.1 | SEC8 | Component of the Exocyst Complex |
| | | Cre04.g224800.t1.1 | | Endosomal R-SNARE protein, Vamp7/Nyv1-family |
| | | Cre17.g728150.t1.1 | | Endosomal R-SNARE protein, Yky6-family |
| | | Cre12.g507450.t1.1 | | Trans-Golgi network Qa-SNARE protein, Syntaxin16/Syx16/Tlg2/Syp4-family |
| | | Cre03.g210600.t1.1 | | NF* |
| | | Cre04.g225900.t1.1 | | Endosomal R-SNARE protein, Vamp7/Nyv1-family |
| | | Cre02.g101400.t1.1 | CHC1 | Clathrin Heavy Chain |
| | | Cre17.g709350.t1.1 | | Late endosomal Qc-SNARE protein, Syx8/Syntaxin8-family |
| | | Cre07.g342050.t1.1 | | Endosomal Qb-SNARE, Npsn-family |
| | | Cre16.g692050.t1.1 | | ER-Golgi Qa-SNARE protein, Syntaxin5/Syx5/Sed5/Syp3-family |
| | | Cre16.g676650.t1.1 | AP1G1 | Gamma1-Adaptin |
| | | Cre02.g099000.t1.1 | | Late endosomal Qc-SNARE protein, Syx6/Tlg1/Syp5/6-family |
| | | Cre12.g554200.t1.2 | | ER-Golgi Qb-SNARE, Memb/GS35/Bos1-family |
| | | Cre06.g310000.t1.1 | AP4E1 | Epsilon4-Adaptin |
| | | Cre10.g421250.t1.1 | EXO70 | Hypothetical Conserved Protein. Similar to Exo70, a subunit of the exocyst complex |
| | | Cre07.g330950.t1.1 | AP4S4 | Sigma4-Adaptin |
| | | Cre12.g488850.t1.2 | | Adaptin, alpha/gamma/epsilon |
| | division | Cre06.g269950.t1.1 | CDC48 | Protein involved in ubiquitin-dependent degradation of ER-bound substrates |
| | | Cre08.g359200.t1.2 | | Regulator of chromosome condensation (RCC1) |
| | organisation | Cre13.g588600.t1.2 | | Kinesin (SMY1 subfamily) |
| | | Cre12.g513450.t1.1 | TUH1 | Eta-Tubulin |
| | | Cre01.g010950.t1.2 | | 26S proteasome regulatory complex, subunit PSMD10 (Ankyrin repeat) |
| | | Cre16.g679650.t1.2 | | Fimbrin/Plastin |
| | | Cre06.g261950.t1.1 | | Myotrophin and similar proteins (Ankyrin repeat) |
| | | Cre06.g291700.t1.1 | RSP3 | Radial spoke protein 3 |
| | | Cre10.g446700.t1.1 | ANK28 | Ankyrin repeat and DHHC-type Zn-finger domain containing proteins |

Table 6. Continued on next page

*Table 6. Continued*

| Primary Mapman class | Secondary Mapman class | Gene ID (v4) | Gene name | Annotation |
|---|---|---|---|---|
| Hormone metabolism† | abscisic acid | Cre16.g657800.t1.2 | CCD3 | Carotenoid cleavage dioxygenase |
| | auxin | Cre14.g609900.t1.1 | | Predicted membrane protein, contains DoH and Cytochrome b-561/ferric reductase transmembrane domains |
| | brassinosteroid | Cre16.g663950.t1.1 | | Sterol C5 desaturase |
| | | Cre02.g092350.t1.1 | | Cytochrome P450, CYP51 superfamily; sterol 14 desaturase |
| | | Cre12.g557900.t1.1 | CDI1 | C-8,7 sterol isomerase |
| | | Cre02.g076800.t1.1 | | Delta14-sterol reductase, mitochondrial |
| | | Cre12.g500500.t1.2 | | 24-methylenesterol C-methyltransferase |
| | ethylene | Cre02.g108450.t1.1 | FAP280 | Flagellar Associated Protein, transcriptional coactivator-like, putative transcription factor |
| | jasmonate | Cre19.g756100.t1.1 | | NF* |
| Misc | acid and other phosphatases | Cre09.g396900.t1.1 | | NADH pyrophosphatase I of the Nudix family of hydrolases |
| | | Cre06.g259650.t1.1 | | Calcineurin-like phosphoesterase, Acid-phosphatase-related |
| | | Cre06.g249800.t1.1 | | Sphingomyelin synthetase -related |
| | cytochrome P450 | Cre05.g234100.t1.1 | | Cytochrome P450, CYP197 superfamily |
| | dynamin | Cre02.g079550.t1.1 | DRP2 | Dynamin-related GTPase, involved in circadian rhythms |
| | | Cre05.g245950.t1.1 | DRP1 | Dynamin-related GTPase |
| | glutathione S transferases | Cre03.g154950.t1.1 | | Glutathione S-transferase |
| | misc2 | Cre12.g538450.t1.1 | EPT1 | CDP-Etn:DAG Ethanolamine phosphotransferase |
| | short chain dehydrogenase/ reductase (SDR) | Cre12.g556750.t1.2 | | Short-chain dehydrogenase/ reductase |
| | | Cre08.g384864.t1.1 | | SH3 domain, protein binding |
| | | Cre27.g775000.t1.1 | | NF* |
| | | Cre17.g731350.t1.2 | | Short chain dehydrogenase |
| | UDP glucosyl and glucoronyl transferases | Cre02.g111150.t1.2 | ELG26 | Exostosin-like glycosyltransferase |
| | | Cre02.g144050.t1.1 | | Acetylglucosaminyltransferase EXT1/exostosin 1 |
| | | Cre03.g204050.t1.2 | ELG6 | Exostosin-like glycosyltransferases |
| | | Cre11.g474450.t1.1 | | NF* |
| | | Cre03.g173300.t1.1 | | Lactosylceramide 4-alpha-galactosyltransferase (alpha-1,4-galactosyltransferase) |
| | | Cre02.g116600.t1.1 | ELG23 | Exostosin-like glycosyltransferase |
| Down-regulated genes | | | | |
| Hormone metabolism† | cytokinin | Cre18.g744950.t1.2 | | NF* |

*Table 6. Continued on next page*

Wakao *et al.* eLife 2014;3:e02286. DOI: 10.7554/eLife.02286

Table 6. Continued

| Primary Mapman class | Secondary Mapman class | Gene ID (v4) | Gene name | Annotation |
|---|---|---|---|---|
| | | Cre16.g678900.t1.1 | | Response regulator receiver domain |
| | | Cre01.g040450.t1.1 | HDT1 | Histidine-aspartic acid phosphotransferase 1 (phosphorylation cascade) |
| | ethylene | Cre09.g403550.t1.1 | | Iron/ascorbate family oxidoreductases |
| Nucleotide metabolism | deoxynucleotide metabolism | Cre12.g491050.t1.1 | RIR2 | Ribonucleotide reductase (RNR), small subunit |
| | | Cre12.g492950.t1.1 | RIR1 | Ribonucleotide reductase (RNR), large subunit, class I |
| | | Cre16.g667850.t1.1 | | dUTP pyrophosphatase |
| | synthesis | Cre14.g614300.t1.1 | | Inosine-5-monophosphate dehydrogenase/GMP reductase |
| | | Cre07.g318750.t1.1 | | Phosphoribosylformylglycinamidine cyclo-ligase |
| Tetrapyrrole synthesis | porphobilinogen deaminase | Cre16.g663900.t1.1 | | Porphobilinogen deaminase |
| | protochlorophyllide reductase | Cre01.g015350.t1.1 | | Light-dependent protochlorophyllide reductase |
| | urogen III methylase | Cre02.g133050.t1.2 | | NF* |
| DNA | repair | Cre16.g670550.t1.2 | | XP-G/RAD2 DNA repair endonuclease |
| | synthesis/chromatin structure | Cre07.g338000.t1.1 | MCM2 | Minichromosome maintenance protein |
| | | Cre07.g314900.t1.2 | | ATP-dependent RNA helicase, DEAD/DEAH helicase |
| | | Cre03.g172950.t1.1 | CBF5 | Centromere/microtubule binding protein |
| | | Cre01.g015250.t1.1 | | Eukaryotic DNA polymerase delta |
| | | Cre27.g774200.t1.2 | | NF* |
| | | Cre07.g316850.t1.1 | MCM4 | Minichromosome maintenance protein |
| | unspecified | Cre10.g451250.t1.2 | | Adenylate and guanylate cyclase catalytic domain, 3-5 exonuclease |
| | | Cre01.g059950.t1.2 | | NF* |

*Corresponding gene model was not found in v5.
†Functional terms are inferred by homology to the annotation set of *Arabidopsis thaliana* (**Lopez et al., 2011**).

By SDS-PAGE and immunoblot analysis, SAK1 appeared in multiple forms with higher molecular weight during acclimation compared to that observed in control cells (*Figures 4F and 6A,C*). When the extracted protein samples were treated with phosphatase, the diffuse pattern of multiple forms collapsed into a single band detected by immunoblot analysis that had an even higher mobility that that of untreated cells (*Figure 6D*). This result indicates that SAK1 is a phosphorylated protein during basal conditions, and it is further phosphorylated upon exposure of cells to $^1O_2$.

## Discussion

### SAK1 is necessary for acclimation of *Chlamydomonas* cells to $^1O_2$

To understand the retrograde signal transduction pathway involved in the cellular response to $^1O_2$, we focused on the unique ability of *Chlamydomonas* to acclimate to $^1O_2$ stress (*Ledford et al., 2007*), and

**Table 7.** Enriched functional classes among differentially expressed genes in *sak1* at basal level

| Primary Mapman class | Secondary Mapman class | Gene ID (v4) | Gene name | Annotation |
|---|---|---|---|---|
| Elevated in *sak1* | | | | |
| nucleotide metabolism | deoxynucleotide metabolism | Cre12.g491050.t1.1 | RIR2 | Ribonucleotide reductase (RNR), small subunit |
| | | Cre12.g492950.t1.1 | RIR1 | Ribonucleotide reductase (RNR), large subunit, class I |
| | | Cre16.g667850.t1.1 | | dUTP pyrophosphatase |
| | phosphotransfer and pyrophosphatases | Cre02.g122450.t1.1 | | NF* |
| | | Cre02.g093950.t1.1 | PYR5 | Uridine 5'- monophosphate synthase/orotate phosphoribosyltransferase |
| | | Cre12.g519950.t1.1 | | Flagellar Associated Protein similar to adenylate/guanylate kinases |
| | | Cre26.g772450.t1.1 | | NF* |
| | synthesis | Cre65.g793400.t1.1 | | NF* |
| | | Cre02.g079700.t1.1 | PYR2 | Aspartate carbamoyltransferase |
| | | Cre01.g048950.t1.1 | | dUTP pyrophosphatase |
| | | Cre07.g318750.t1.1 | | Phosphoribosylformylglycinamidine cyclo-ligase. |
| DNA | repair | Cre07.g314650.t1.1 | | Chloroplast RecA recombination protein |
| | synthesis/chromatin structure | Cre04.g214350.t1.2 | | Eukaryotic DNA polymerase alpha, catalytic subunit |
| | | Cre07.g314900.t1.2 | | ATP-dependent RNA helicase (DEAD/DEAH) |
| | | Cre04.g223850.t1.1 | | Cytoplasmic DExD/H-box RNA helicase |
| | | Cre01.g015250.t1.1 | | Eukaryotic DNA polymerase delta, catalytic subunit. |
| | | Cre07.g342506.t1.1 | | Ubiquitin-protein ligase |
| | | Cre07.g338000.t1.1 | MCM2 | Minichromosome maintenance protein |
| | | Cre03.g178650.t1.1 | MCM6 | MCM6 DNA replication protein |
| | | Cre07.g312350.t1.2 | | DNA polymerase alpha, primase subunit |
| | | Cre01.g009250.t1.2 | TOP2 | DNA topoisomerase II |
| | | Cre26.g772150.t1.1 | | NF* |
| | | Cre07.g316850.t1.1 | MCM4 | Minichromosome maintenance protein 4 |
| | | Cre06.g263800.t1.2 | | tRNA-splicing endonuclease positive effector (SEN1) |
| | | Cre06.g295700.t1.2 | MCM3 | Minichromosome maintenance protein |
| | | Cre06.g251800.t1.1 | RFC4 | DNA replication factor C complex subunit 4 |
| | unspecified | Cre07.g322300.t1.2 | | DNA repair helicase of the DEAD superfamily |
| | | Cre17.g718100.t1.1 | | Phosphatidylinositol transfer protein SEC14 and related proteins (CRAL/TRIO) |

*Table 7. Continued on next page*

*Table 7. Continued*

| Primary Mapman class | Secondary Mapman class | Gene ID (v4) | Gene name | Annotation |
|---|---|---|---|---|
| Tetrapyrrole synthesis | Glu-tRNA reductase | Cre07.g342150.t1.1 | HEM1 | Glutamyl-tRNA reductase |
| | Glu-tRNA synthetase | Cre44.g788000.t1.1 | | Glutamyl-tRNA reductase |
| | | Cre06.g306300.t1.1 | CHLI1 | Magnesium chelatase subunit I |
| | magnesium chelatase | Cre07.g325500.t1.1 | | Magnesium chelatase subunit H |
| | protochlorophyllide reductase | Cre01.g015350.t1.1 | POR1 | Light-dependent protochlorophyllide reductase |
| Photosynthesis | Calvin-Benson cycle | Cre05.g234550.t1.1 | | Fructose-biphosphate aldolase |
| | light reaction | Cre07.g330250.t1.1 | PSAH | Subunit H of photosystem I |
| | | Cre07.g334550.t1.1 | | Photosystem I subunit PsaO |
| | | Cre06.g261000.t1.1 | PSBR | 10 kDa photosystem II polypeptide |
| | photorespiration | Cre12.g542300.t1.1 | GYK1 | Glycerate kinase |
| | | Cre06.g253350.t1.1 | GCSH | Glycine cleavage system, H-protein |
| | | Cre06.g293950.t1.1 | SHMT2 | Serine hydroxymethyltransferase 2 |
| Transport | ABC transporters and multidrug resistance systems | Cre04.g222700.t1.1 | | ATPase component of ABC transporters with duplicated ATPase domains/Translation elongation factor EF-3b |
| | | Cre17.g728400.t1.2 | | ABCtransporter (ABC-2 type) |
| | | Cre05.g241350.t1.2 | | ABCtransporter (ABC-2 type) |
| | | Cre03.g169300.t1.1 | | ABCtransporter (ABC-2 type) |
| | | Cre11.g474600.t1.1 | | NF* |
| | amino acids | Cre04.g226150.t1.2 | AOC1 | Amino acid carrier 1; belongs to APC (Amino acid Polyamine organo Cation) family |
| | calcium | Cre09.g388850.t1.1 | ACA1 | P-type ATPase/cation transporter, plasma membrane |
| | metabolite transporters at the envelope membrane | Cre06.g263850.t1.2 | TPT2 | Triose phosphate/phosphate translocator |
| | metabolite transporters at the mitochondrial membrane | Cre10.g449100.t1.1 | | Mitochondrial oxodicarboxylate carrier protein |
| | | Cre01.g069350.t1.1 | | NF* |
| | | Cre15.g641200.t1.1 | | Mitochondrial fatty acid anion carrier protein/Uncoupling protein |
| | | Cre09.g396350.t1.1 | | Mitochondrial carrier protein PET8 |
| | misc | Cre06.g311000.t1.2 | FBT2 | Folate transporte |
| | | Cre17.g718100.t1.1 | | Phosphatidylinositol transfer protein SEC14 and related proteins (CRAL/TRIO) |
| | phosphate | Cre16.g686750.t1.1 | PTA3 | Proton/phosphate symporter |
| | | Cre16.g675300.t1.2 | | Sodium-dependent phosphate transporter, major facilitator superfamily |
| | potassium | Cre12.g553450.t1.2 | | NF* |
| | sulphate | Cre17.g723350.t1.1 | SUL2 | Sulfate anion transporter |
| | unspecified cations | Cre13.g573900.t1.1 | | Na+:iodide/myo-inositol/ multivitamin symporters |

*Table 7. Continued on next page*

*Table 7. Continued*

| Primary Mapman class | Secondary Mapman class | Gene ID (v4) | Gene name | Annotation |
|---|---|---|---|---|
| | sugars | Cre16.g675300.t1.2 | | Sodium-dependent phosphate transporter, major facilitator superfamily |
| RNA | processing | Cre10.g427700.t1.1 | | ATP-dependent RNA helicase, DEAD/DEAH box helicase |
| | | Cre12.g538750.t1.1 | LSM1 | U6 snRNA-associated Sm-like protein LSm1, RNA cap binding; (SMP6d) |
| | | Cre10.g433750.t1.2 | PAP1 | Nuclear poly(A) polymerase |
| | | Cre03.g182950.t1.1 | | NF* |
| | | Cre08.g375128.t1.1 | | NF* |
| | regulation of transcription | Cre17.g728200.t1.2 | | YL-1 protein (transcription factor-like 1) |
| | | Cre06.g275500.t1.1 | | AP2 Transcription factor |
| | | Cre28.g777500.t1.2 | | NF* |
| | | Cre13.g572450.t1.1 | | Response regulator receiver domain (sensor histidine kinase-related, regulation of transcription) |
| | | Cre14.g620500.t1.1 | | AP2 Transcription factor |
| | | Cre16.g673150.t1.1 | | Histone deacetylase complex, catalytic component RPD3 |
| | | Cre02.g078700.t1.2 | | DNA damage-responsive repressor GIS1/RPH1, jumonji superfamily |
| | | Cre03.g198800.t1.1 | | Myb-like DNA-binding domain |
| | | Cre04.g218050.t1.2 | | RWP-RK domain |
| | | Cre07.g324400.t1.1 | VPS24 | Subunit of the ESCRT-III complex, vacuoloar sortin protein |
| | | Cre11.g481050.t1.1 | | SWI/SNF-related chromatin binding protein |
| | | Cre02.g101950.t1.1 | TMU2 | tRNA (uracil-5)-methyltransferase |
| | | Cre10.g459600.t1.2 | | CAATT-binding transcription factor/60S ribosomal subunit biogenesis protein |
| | | Cre01.g018650.t1.2 | | NF* |
| | | Cre01.g012200.t1.2 | | NF* |
| | | Cre02.g129750.t1.1 | | NF* |
| | | Cre10.g461750.t1.2 | | DNA (cytosine-5-)-methyltransferase |
| | | Cre01.g004600.t1.2 | RWP12 | Putative RWP-RK domain transcription factor |
| | | Cre09.g400100.t1.1 | | Predicted Zn-finger protein, zinc and DNA binding domains |
| | | Cre07.g335150.t1.2 | | SBP domain |
| | RNA binding | Cre16.g662700.t1.1 | | NF* |
| | | Cre07.g330300.t1.1 | | RNA-binding protein musashi/mRNA cleavage and polyadenylation factor I complex, subunit HRP1 |

*Table 7. Continued on next page*

*Table 7. Continued*

| Primary Mapman class | Secondary Mapman class | Gene ID (v4) | Gene name | Annotation |
|---|---|---|---|---|
| | | Cre06.g275100.t1.1 | | RNA-binding protein musashi/ mRNA cleavage and polyadenylation factor I complex, subunit HRP1 |
| | transcription | Cre07.g322200.t1.1 | | NF* |
| **Repressed in *sak1*** | | | | |
| Transport | ABC transporters and multidrug resistance systems | Cre02.g097800.t1.2 | | ABC transporter (MDR) |
| | | Cre17.g725200.t1.1 | | ABC transporter, peptide exporter |
| | | Cre13.g580300.t1.1 | | ABC transporter family protein |
| | | Cre10.g439000.t1.2 | | Long-chain acyl-CoA transporter, ABC superfamily (involved in peroxisome organization and biogenesis) |
| | amino acids | Cre06.g292350.t1.1 | AOC4 | Amino acid carrier |
| | calcium | Cre06.g263950.t1.2 | | Sodium/potassium-transporting ATPase subunit alpha |
| | | Cre16.g681750.t1.2 | | Calcium transporting ATPase |
| | metabolite transporters at the mitochondrial membrane | Cre03.g172300.t1.1 | | Mitochondrial phosphate carrier protein |
| | | Cre09.g394800.t1.2 | | Mitochondrial substrate carrier protein |
| | metal | Cre03.g189550.t1.2 | ZIP3 | Zinc transporter, ZIP family |
| | | Cre11.g479600.t1.2 | | Sodium/calcium exchanger NCX1 and related proteins |
| | | Cre06.g281900.t1.1 | ZIP7 | Zinc transporter and related ZIP domain-containing proteins |
| | misc | Cre02.g089900.t1.1 | | Secretory carrier membrane protein |
| | | Cre10.g448050.t1.1 | | Retinaldehyde binding protein-related (CRAL/TRIO domain) |
| | | Cre03.g177750.t1.2 | | Multidrug resistance pump |
| | NDP-sugars at the ER | Cre02.g112900.t1.1 | | GDP-fucose transporter (Triose-phosphate transporter family) |
| | P- and V-ATPases | Cre01.g027800.t1.1 | ATPvH | Vacuolar ATP synthase subunit H |
| | | Cre10.g446550.t1.1 | ATPvF | Vacuolar ATP synthase subunit F |
| | | Cre03.g176250.t1.1 | ATPvD1 | Vacuolar ATP synthase subunit D |
| | | Cre06.g250250.t1.1 | ATPvC | Vacuolar ATP synthase subunit C |
| | | Cre10.g459200.t1.1 | ACA4 | P-type ATPase/cation transporter, plasma membrane (Low CO2 inducible gene) |
| | phosphate | Cre12.g515750.t1.2 | | Sodium-dependent phosphate transporter-related |
| | | Cre08.g379550.t1.2 | | Sodium-dependent phosphate transporter, major facilitator superfamily |
| | | Cre12.g489400.t1.1 | PTB7 | Putative phosphate transporter, sodium/phosphate transporter |

*Table 7. Continued on next page*

*Table 7. Continued*

| Primary Mapman class | Secondary Mapman class | Gene ID (v4) | Gene name | Annotation |
|---|---|---|---|---|
| | | Cre02.g144650.t1.1 | PTB12 | Sodium/phosphate symporter |
| | unspecified anions | Cre09.g404100.t1.1 | | Cl- channel CLC-7 and related proteins (CLC superfamily) |
| | | Cre17.g729450.t1.1 | | Cl- channel CLC-7 and related proteins (CLC superfamily) |
| | | Cre01.g037150.t1.2 | | Voltage-gated chloride channel activity |
| | sugars | Cre03.g206800.t1.2 | HXT1 | Hexose transporter |
| | P- and V-ATPases | Cre03.g176250.t1.1 | ATPvD1 | Vacuolar ATP synthase subunit D |
| | | Cre10.g446550.t1.1 | ATPvF | Vacuolar ATP synthase subunit F |
| | | Cre01.g027800.t1.1 | ATPvH | Vacuolar ATP synthase subunit H |
| Mitochondrial electron transport / ATP synthesis | cytochrome c reductase | Cre01.g051900.t1.1 | RIP1 | Rieske iron-sulfur protein of mitochondrial ubiquinol-cytochrome c reductase (complex III) |
| | | Cre06.g262700.t1.2 | | Ubiquinol cytochrome c reductase, subunit 7 |
| | $F_1$-ATPase | Cre02.g116750.t1.2 | | F0F1-type ATP synthase, alpha subunit |
| | | Cre01.g018800.t1.1 | ATP6 | Mitochondrial F1F0 ATP synthase subunit 6 |
| | | Cre10.g420700.t1.1 | | Mitochondrial F1F0-ATP synthase, subunit epsilon/ATP15 |
| | | Cre16.g680000.t1.1 | ATP5 | Mitochondrial ATP synthase subunit 5, OSCP subunit |
| | NADH-DH | Cre10.g434450.t1.1 | NUOA9 | Putative NADH:ubiquinone oxidoreductase (Complex I) 39 kDa subunit |
| | | Cre08.g378900.t1.1 | NUO3 | NADH:ubiquinone oxidoreductase ND3 subunit |
| | | Cre10.g450400.t1.1 | NUO5 | NADH:ubiquinone oxidoreductase (Complex I) 24 kD subunit |
| Lipid metabolism | 'exotics' (steroids, squalene etc) | Cre14.g615050.t1.1 | | 3-oxo-5-alpha-steroid 4-dehydrogenase, Steroid reductase required for elongation of the VLCFAs (enoyl reductase) |
| | | Cre12.g530550.t1.2 | KDG2 | Diacylglycerol kinase, sphingosine kinase |
| | | Cre02.g137850.t1.1 | | NF* |
| | FA desaturation | Cre17.g711150.t1.1 | | Omega-6 fatty acid desaturase (delta-12 desaturase) |
| | glyceral metabolism | Cre13.g577450.t1.2 | | Glycerol-3-phosphate dehydrogenase |
| | glycolipid synthesis | Cre13.g583600.t1.1 | DGD1 | Digalactosyldiacylglycerol synthase |
| | | Cre16.g656400.t1.1 | SQD1 | UDP-sulfoquinovose synthase |
| | lipid degradation | Cre06.g252801.t1.2 | | CGI-141-related/lipase containing protein (TAG lipase) |
| | | Cre03.g164350.t1.2 | | Lysophospholipase, putative drug exporter of the RND superfamily |

*Table 7. Continued*

| Primary Mapman class | Secondary Mapman class | Gene ID (v4) | Gene name | Annotation |
|---|---|---|---|---|
| | phospholipid synthesis | Cre06.g281250.t1.1 | CFA1 | Cyclopropane fatty acid synthase |
| | | Cre09.g398700.t1.1 | CFA2 | Cyclopropane fatty acid synthase |
| | | Cre11.g472700.t1.1 | | NF* |
| | | Cre06.g262550.t1.1 | | Zinc finger MYND domain containing protein 10 |
| Photosynthesis | Calvin-Benson cycle | Cre12.g511900.t1.1 | RPE1 | Ribulose phosphate-3-epimerase |
| | | Cre02.g120100.t1.1 | RBCS1 | Ribulose-1,5-bisphosphate carboxylase/oxygenase small subunit 1 |
| | light reaction | Cre05.g243800.t1.1 | CPLD45 | Photosystem II Psb27 protein |
| | | Cre10.g420350.t1.1 | PSAE | Photosystem I reaction center subunit IV |
| | | Cre01.g071450.t1.2 | | NF* |
| | | Cre06.g291650.t1.1 | | Ferredoxin |
| | | Cre05.g242400.t1.1 | | No functional annotation |
| | photorespiration | Cre09.g411900.t1.2 | SHMT3 | Serine hydroxymethyltransferase 3 |
| | | Cre06.g295450.t1.1 | HPR1 | Hydroxypyruvate reductase |
| Major CHO metabolism | degradation | Cre09.g415600.t1.2 | | Starch binding domain |
| | | Cre11.g473500.t1.2 | | NF* |
| | | Cre09.g415600.t1.2 | | Starch binding domain |
| | synthesis | Cre06.g289850.t1.2 | SBE1 | Starch Branching Enzyme |
| | | Cre17.g721500.t1.1 | | Granule-bound starch synthase I |
| misc | acid and other phosphatases | Cre13.g568600.t1.2 | | Multiple inositol polyphosphate phosphatase-related, Acid phosphatase activity |
| | alcohol dehydrogenases | Cre13.g569350.t1.1 | | Sterol dehydrogenase-related, Flavonol reductase/cinnamoyl-CoA reductase |
| | cytochrome P450 | Cre07.g356250.t1.2 | | Cytochrome P450 CYP4/CYP19/CYP26 subfamilies, beta-carotene 15,15'-monooxygenase |
| | | Cre07.g356250.t1.2 | | Cytochrome P450 CYP4/CYP19/CYP26 subfamilies, beta-carotene 15,15'-monooxygenase |
| | dynamin | Cre17.g724150.t1.1 | DRP3 | Dynamin-related GTPase |
| | GCN5-related N-acetyltransferase | Cre16.g657150.t1.2 | | N-acetyltransferase activity (GNAT) family |
| | gluco-, galacto- and mannosidases | Cre03.g171050.t1.2 | GHL1 | Glycosyl hydrolase |
| | misc2 | Cre14.g614100.t1.1 | GTR26 | Dolichyl-diphosphooligosaccharide-protein glycosyltransferase |
| | rhodanese | Cre07.g352550.t1.1 | RDP3 | Putative rhodanese domain phosphatase |
| | short chain dehydrogenase/reductase (SDR) | Cre07.g352450.t1.1 | | Corticosteroid 11-beta-dehydrogenase and related short chain-type dehydrogenases, 3-hydroxybutyrate dehydrogenase |

*Table 7. Continued on next page*

Table 7. Continued

| Primary Mapman class | Secondary Mapman class | Gene ID (v4) | Gene name | Annotation |
|---|---|---|---|---|
| | | Cre12.g559350.t1.1 | | 1-Acyl dihydroxyacetone phosphate reductase and related dehydrogenases |
| | | Cre03.g191850.t1.1 | | Short chain dehydrogenase |
| | UDP glucosyl and glucoronyl transferases | Cre11.g474450.t1.1 | | NF* |
| | | Cre03.g205250.t1.2 | ELG4 | Exostosin-like glycosyltransferase |
| | | Cre16.g659500.t1.1 | | Lactosylceramide 4-alpha-galactosyltransferase |
| | | Cre11.g483400.t1.2 | ELG10 | Exostosin-like glycosyltransferase |
| Tetrapyrrole synthesis | Glu-tRNA synthetase | Cre12.g510800.t1.1 | CHLI2 | Magnesium-chelatase subunit chlI |
| | magnesium protoporphyrin IX methyltransferase | Cre12.g498550.t1.2 | | Magnesium protoporphyrin IX S-adenosyl methionine O-methyl transferase (Magnesium-protoporphyrin IX methyltransferase) (PPMT) |
| | unspecified | Cre12.g516350.t1.1 | COX10 | Cytochrome c oxidase assembly protein Cox10 |
| | urogen III methylase | Cre02.g133050.t1.2 | | NF* |

*Corresponding gene model was not found in v5.

we isolated a regulatory mutant that is unable to acclimate. Several previous genetic screens aimed at dissecting the mechanisms of $^1O_2$ signaling have concentrated on the nuclear gene expression response to $^1O_2$, often relying on the response of a single marker gene (*Baruah et al., 2009a*; *Brzezowski et al., 2012*; *Fischer et al., 2012*; *Shao et al., 2013*). In contrast, our screen exploited a physiological response to sublethal levels of $^1O_2$, which induces the wild type to survive a subsequent, otherwise lethal treatment with the $^1O_2$ generator RB (*Ledford et al., 2007*). The *sak1* mutant completely lacks this ability to acclimate to $^1O_2$ (*Figure 1A*). An analogous phenotype is exhibited by the *yap1Δ* mutant of *Saccharomyces cerevisiae*, which is unable to acclimate to hydrogen peroxide stress (*Stephen et al., 1995*).

In contrast to the complete loss of acclimation to RB, *sak1* acclimates (but less effectively than WT) when pretreated with high light and challenged with RB (*Figure 1B*). This result suggests that the high light pretreatment induces a broader response than that elicited by RB and that *sak1* is still able to respond to other signals besides $^1O_2$ (e.g., plastoquinone redox state, $H_2O_2$, and/or superoxide) that are involved in the response to high light. When tested on TAP agar plates for photoheterotrophic growth in the presence of various photosynthetic inhibitors, the *sak1* mutant displayed sensitivity to RB but not to other inhibitors (*Figure 1D*). In particular, *sak1* is not more sensitive than WT to high light or norflurazon (an inhibitor of the biosynthesis of carotenoids, which function as quenchers of $^1O_2$). We speculate that the lack of $^1O_2$-sensitive phenotype in these plate experiments is attributable to the time-scale of the treatments involved. $^1O_2$ generated by RB or during a transfer to higher light intensity is transient, whereas NF requires longer time to exert its effect because it needs to enter the cell, inhibit biosynthesis, and deplete cells of existing carotenoids. During this time, the cell is likely able to acclimate by detoxifying and reducing the generation of $^1O_2$ by various means such as changing the composition of the photosynthetic apparatus. We have previously shown that acclimation to $^1O_2$ is transient and is dissipated by 24 hr post-treatment (*Ledford et al., 2007*). Consistent with this, pretreatment with RB does not acclimate the cells to stresses such as growth in high light or norflurazon that require a period of days to assess an effect on viability (*Figure 1—figure supplement 1*). We have also observed that under our experimental conditions, the induction of target gene expression upon exposure to $^1O_2$ lasts up to 90 min and then declines. We conclude that SAK1 functions mainly during transient perturbations that generate $^1O_2$. However, during steady-state growth under high light or norflurazon, the cell is able to cope by other means that do not involve *SAK1*.

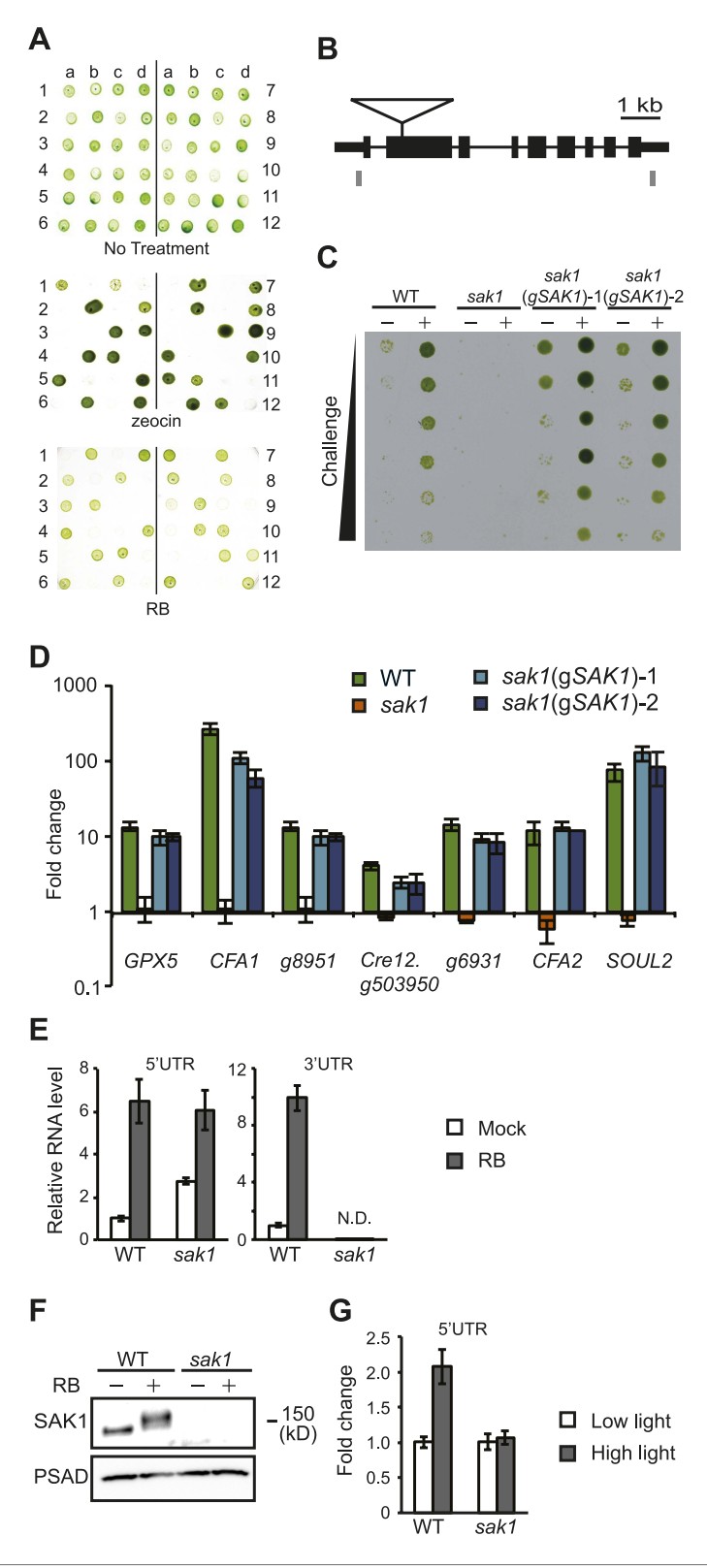

**Figure 4**. Genetic and molecular analysis of *sak1*. (**A**) The insertion of a zeocin resistance gene and the RB sensitivity phenotype are linked. Twelve complete tetrads from a backcross of *sak1* to wild type are shown. Numbers indicate independent tetrads, and letters (a-d) indicate the individual progeny from tetrads. (**B**) Gene
*Figure 4. Continued on next page*

*Figure 4. Continued*

structure of *SAK1* and the insertion site. Gray boxes indicate positions of primers used for qPCR. (**C**) Transformation of *sak1* with a genomic fragment containing *SAK1* rescues the acclimation phenotype. *sak1*(g*SAK1*)-1 and *sak1*(g*SAK1*)-2 are two independent transformants. (**D**) *sak1*(g*SAK1*)-1 and *sak1*(g*SAK1*)-2 show recovery of $^1O_2$ target gene expression. Y-axis indicates fold change during acclimation to $^1O_2$. (**E**) qRT-PCR of *SAK1* in WT and *sak1* mutant using primers for 5'- and 3'-UTR shown in panel **B**. (**F**) SAK1 protein is induced in WT and detected as higher molecular weight bands during acclimation to $^1O_2$ generated by RB. (**G**) *SAK1* transcript probed for 5'-UTR in cells transferred from low light to high light for 1 hr. Error bars indicate standard deviation of biological triplicates.

## SAK1 is necessary for a subset of the genome-wide response to $^1O_2$ in *Chlamydomonas*

A physiological acclimation response that results in such an evident growth phenotype (*Figure 1A*) likely involves large-scale changes in gene expression, and transcriptome analysis of wild-type cells showed that hundreds of nuclear genes are up- or down-regulated during acclimation to $^1O_2$ (*Figure 3A,B*; *Supplementary file 1*, C1). The *sak1* mutant is specifically impaired in regulation of a notable subset of these genes, that is, those that are most strongly induced in the wild type (*Table 5*), suggesting that these genes play a key role in the acclimation response to $^1O_2$.

In particular, many genes involved in sterol and lipid metabolism were induced by $^1O_2$ in *Chlamydomonas* (*Figure 3B*; *Table 3*). For example, two genes encoding putative cyclopropane fatty acid synthase (*CFA1* and *CFA2*) exhibited SAK1-dependent induction (*Figure 2*). Cyclopropane fatty acids have been found in large amounts in the seeds of *Sterculia foetida* (*Bao et al., 2002*), although its biological function is unknown. In bacteria, it has been implicated in oxidative stress responses (*Guerzoni et al., 2001*; *Kim et al., 2005*) and particularly in the anoxygenic photosynthetic bacterium *Rhodobacter sphaeroides,* CFA gene expression is induced during $^1O_2$ stress by a σ$^E$ factor (*Ziegelhoffer and Donohue, 2009*). Interestingly CFA mutants of *R. sphaeroides* are compromised in the induction of genes in response to $^1O_2$, suggesting a regulatory role of the gene, protein, or the product of its enzymatic function (cyclopropane fatty acids, *Bao et al., 2002*) in gene expression rather than solely a biochemical stress response (*Nam et al., 2013*).

Another intriguing class of up-regulated genes enriched during $^1O_2$ acclimation in WT and not in *sak1* was a group of genes encoding transporters, especially ABC transporters related to the MDR and PDR types. This was not surprising considering that $^1O_2$ exists in aquatic and terrestrial environments, where it is generated by photosensitizing humic substances (*Frimmel et al., 1987*; *Steinberg et al., 2008*), which are known to affect microbial populations including phytoplankton (*Glaeser et al., 2010*, *2014*). Assuming that some of these transporters function to export photosensitizing molecules from the cell, our results suggest that removal of photosensitizers is an integral part of the $^1O_2$ response in *Chlamydomonas*, rather than simply a response to the presence of a xenobiotic compound such as RB (*Table 4*). It is likely that *Chlamydomonas*, a soil-dwelling microalga, needs to respond to $^1O_2$ that is generated not only in the chloroplast, but also in other compartments. In this context, it is noteworthy that a recent study has demonstrated light-independent $^1O_2$ generation in multiple organelles other than the chloroplast under various biotic and abiotic stresses in plants (*Mor et al., 2014*).

Two proteins with SOUL heme-binding domains were among SAK1-dependent up-regulated genes (SOUL2 and Cre06.g299700.t1.1, formerly annotated as SOUL1 in v4). Aside from their ability to bind various porphyrins (*Blackmon et al., 2002*; *Sato et al., 2004*), SOUL heme-binding proteins have been described in diverse biological functions in mice, such as in apoptosis by interacting with a mitochondrial anti-apoptotic factor Bcl-xL (*Ambrosi et al., 2011*) or an isoform-specific role in retina and pineal gland (*Zylka and Reppert, 1999*). The latter form is suggested to play a role in transporting heme or by binding free heme to prevent oxidative stress (*Sato et al., 2004*). In *Arabidopsis* a chloroplast-localized SOUL5 protein has been shown to interact with a heme oxygenase, HY1, and mutation of the gene encoding SOUL5 causes oxidative stress (*Lee et al., 2012*). *Chlamydomonas* contains five putative SOUL heme-binding proteins, only one of which contains an amino-terminal chloroplast transit peptide. The two SOUL protein genes induced by $^1O_2$ in our study do not seem to be targeted to the chloroplast, and they may function in the cytosol where SAK1 resides. It would be interesting to test whether these proteins bind porphyrins and are required for $^1O_2$ acclimation.

**Table 8.** SAK1 domain containing proteins in chlorophytes

| Number in alignment | Organism | Transcript/Protein IDaTranscript/Protein IDaTranscript/Protein ID* |
|---|---|---|
| 1 | *Volvox carteri* | Vocar20009235 |
| 2 | *Volvox carteri* | Vocar20002437 |
| 3 | *Volvox carteri* | Vocar20002672 |
| 4 | *Volvox carteri* | Vocar20004923 |
| 5 | *Volvox carteri* | Vocar20012349 |
| 6 | *Volvox carteri* | Vocar20005988 |
| 7 | *Volvox carteri* | Vocar20007158 |
| 8 | *Volvox carteri* | Vocar20007883 |
| 9 | *Coccomyxa subellipsoidea* | 57405 |
| 10 | *Coccomyxa subellipsoidea* | 59655 |
| 11 | *Coccomyxa subellipsoidea* | 57694 |
| 12 | *Chlamydomonas reinhardtii* | Cre16.g652650.t1.3 |
| 13 | *Chlamydomonas reinhardtii* | Cre06.g271000.t1.2 |
| 14 | *Chlamydomonas reinhardtii* | Cre06.g285800.t1.2 |
| 15 | *Chlamydomonas reinhardtii* | Cre06.g275600.t1.2 |
| 16 | *Chlamydomonas reinhardtii* | Cre06.g285750.t1.3 |
| 17 | *Chlamydomonas reinhardtii* | Cre06.g270950.t1.2 |
| 18 | *Chlamydomonas reinhardtii* | g9774.t1 |
| SAK1 | *Chlamydomonas reinhardtii* | KF985242 |
| 20 | *Chlamydomonas reinhardtii* | Cre03.g179150.t1.2 |
| 21 | *Chlamydomonas reinhardtii* | g3701.t1 |
| 22 | *Chlamydomonas reinhardtii* | Cre03.g179250.t1.2 |
| 23 | *Chlamydomonas reinhardtii* | Cre03.g179200.t1.2 |
| 24 | *Chlamydomonas reinhardtii* | Cre01.g004800.t1.2 |
| 25 | *Chlamydomonas reinhardtii* | Cre01.g048550.t1.3 |
| 26 | *Chlorella variabilis* | EFN51260 |
| 27 | *Chlorella variabilis* | EFN53496 |
| 28 | *Chlorella variabilis* | EFN55618 |

*Table 8. Continued on next page*

A recent study reported the role of bilins in retrograde signaling in *Chlamydomonas* through characterization of heme oxygenase mutants disrupted in bilin biosynthesis and transcriptome analyses during dark to light transitions (**Duanmu et al., 2013**). The transcriptome changes indicated that much of the cell's response during a dark-to-light transition (DL) involves photo-oxidative stress. Interestingly, among the 515 genes up-regulated in WT during $^1O_2$ acclimation, 144 genes overlapped with those that are induced during DL (**Table 9**). Focusing on the 104 genes that we defined as SAK1-dependent (**Table 5**), 31 genes overlapped (**Table 9**). *CFA1*, *CFA2*, and *SOUL2* were among these genes, suggesting that a part of the gene expression response to DL in *Chlamydomonas* is a response to $^1O_2$. *SAK1* itself was also up-regulated during DL as was *SOR1*, which encodes a more broadly oxidative stress-responsive bZIP transcription factor (**Fischer et al., 2012**). We found that 64 of the genes induced during acclimation to $^1O_2$ were also up-regulated in the gain-of-function *sor1* mutant (**Fischer et al., 2012**). However, the most strongly induced SAK1-dependent genes were not among these genes, except for *GPX5*, consistent with the idea that *SAK1* and *SOR1* function in different pathways.

## SAK1 is a key intermediate component in the retrograde signaling pathway for $^1O_2$ acclimation

Cloning of the *SAK1* gene revealed that it encodes a large previously uncharacterized phosphoprotein located primarily in the cytosol (**Figure 6A,D**), suggesting that it functions as an intermediate in the retrograde signaling pathway from the chloroplast to the nucleus that leads to $^1O_2$ acclimation. Previous genetic screens in *Arabidopsis* have identified proteins in the chloroplast, such as EX1 and EX2 (**Wagner et al., 2004**; **Lee et al., 2007**), and in the nucleus, such as PLEIOTROPIC RESPONSE LOCUS 1 (**Baruah et al., 2009b**) and topoisomerase VI (**Simková et al., 2012**), that are involved in $^1O_2$ signaling. By screening for mutants that are unable to induce a $^1O_2$-responsive reporter gene (*HPS70A*) in *Chlamydomonas*, a small zinc finger protein (Cre09.g416500.t1.2) called MBS was recently identified as having a role in ROS signaling in both *Chlamydomonas* and *Arabidopsis* (**Shao et al., 2013**). Like SAK1, MBS in *Chlamydomonas* is located in the cytosol, raising a question about the relationship of these two proteins in $^1O_2$ signaling. As expected, we found *HSP70A* among the genes induced by RB

*Table 8. Continued*

| Number in alignment | Organism | Transcript/Protein IDaTranscript/Protein IDaTranscript/Protein ID* |
|---|---|---|
| 29 | *Chlorella variabilis* | EFN57652 |
| 30 | *Chlorella variabilis* | EFN55658 |
| 31 | *Chlorella variabilis* | EFN54262 |
| 32 | *Chlorella variabilis* | EFN54510 |
| 33 | *Chlorella variabilis* | EFN55806 |
| 34 | *Chlorella variabilis* | EFN53492 |
| 35 | *Micromonas sp. RCC299* | ACO61347 |
| 36 | *Micromonas pusilla CCMP1545* | EEH57791 |
| 37 | *Micromonas sp. RCC299* | ACO65814 |

*1–25, as defined on phytozome.net; 26–37, CrSAK1, genbank accession numbers.

treatment of *Chlamydomonas* (*Table 3*) however in *sak1* it was not significantly induced above the twofold threshold, suggesting that SAK1 might function in the same signaling pathway as MBS. The *MBS* gene itself is not induced by $^1O_2$ (*Shao et al., 2013*), and we will investigate the genetic and biochemical relationship of SAK1 and MBS in future research.

SAK1 contains a novel domain of ~150 amino acid residues that is found in several chlorophyte species (*Table 8*). The sequence of this domain is not highly conserved (*Figure 5—figure supplement 1*), and is even less conserved among land plant proteins, although it is detectable by PSI-BLAST, indicating that it has diverged in sequence in plants and algae. We identified 37 proteins that have the SAK1 domain, 13 of which also contained a bZIP transcription factor domain, consistent with a function in regulating gene expression. Under our standard laboratory growth conditions, SAK1 appears to have a relatively low level of phosphorylation, but it becomes hyperphosphorylated during $^1O_2$ acclimation (*Figure 6D*). Phosphorylation prediction software NetPhos 2.0 (http://www.cbs.dtu.dk/services/NetPhos/) predicted 24 serine, 9 threonine, and one tyrosine residue as possible sites throughout the protein (*Figure 5—figure supplement 3*). One of these serine residues is within the conserved SAK1 domain and is relatively conserved for polar amino acids. At this position, 18 SAK1 family members had threonine, and three had serine residues including SAK1 (*Figure 5—figure supplement 1*). We speculate that phosphorylation of SAK1 in the cytosol is a necessary intermediate step in $^1O_2$ acclimation. Through further analysis of the transcriptome data, isolation of proteins that physically interact with SAK1, and characterization of additional, non-allelic *sak* mutants, we hope to identify the kinase that is responsible for the direct modification of SAK1 as well as other upstream and downstream components of this retrograde signaling pathway in *Chlamydomonas*.

## Material and methods

### *Chlamydomonas* strains and culture conditions

The *sak1* mutant was generated by insertional mutagenesis as described previously (*Dent et al., 2005*) from WT strain 4A+. Cells were grown at 22°C photoheterotrophically in Tris-acetate phosphate media (TAP) unless otherwise stated (*Harris, 2009*).

### RB sensitivity screen and acclimation assays

For systematic screening of large number of strains for increased or decreased resistance to RB, individual strains were inoculated into 180-200 µl TAP medium in 96-well plates, grown for a at least 3 days to saturation under light intensity of 60–80 µmol photons m$^{-2}$ s$^{-1}$, spotted onto TAP plates with 2.7, 3.0, or 3.3 µM RB, and scored for their growth compared to WT and *sak1*. For more quantitative evaluation of RB sensitivity, the cells were grown to saturation in 1 ml of TAP medium because we have observed rapidly growing cells to have more variable sensitivity to RB (data not shown). The cells were counted and adjusted to equal cell density then dispensed into aliquots in duplicate 96-well plates. One of the duplicates was pretreated in dark while the other was placed in light for 40 min with 1 µM RB. For challenge treatments, 4.5, 5.1, 5.7, 6.3, 6.9, and 7.5 µM RB was added to both plates, which were placed under light for 1 hr and then spotted onto TAP agar media with no RB. All treatments were applied under light intensity of 60–80 µmol photons m$^{-2}$ s$^{-1}$, which is the light intensity described as low light unless stated otherwise.

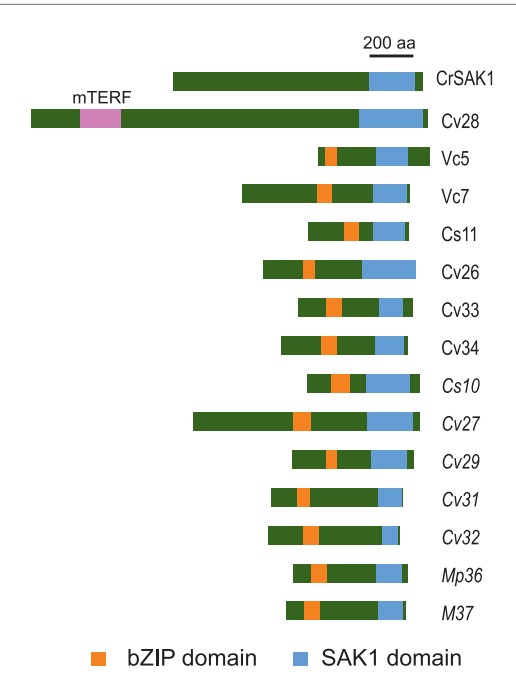

**Figure 5**. SAK1 contains an uncharacterized domain present in some bZIP transcription factors. Schematic of relative positions of SAK1 and bZIP domains. One protein (Cv28) contains a mitochondrial termination factor (mTERF) domain. The letters and numbers in the abbreviated names represent initials of the species and numbers listed in **Table 8**. Proteins with italicized names contain bZIP domains that were recognized by Pfam but scored below significance.

The following figure supplements are available for figure 5:

**Figure supplement 1**. Multiple sequence alignment of SAK1 domains.

**Figure supplement 2**. Secondary structure prediction of SAK1 domain.

**Figure supplement 3**. Prediction of phosphorylation sites in SAK1.

## Pretreatment and challenge with RB and $F_v/F_m$ measurement

Cells were grown under 100 μmol photons m$^{-2}$ s$^{-1}$, adjusted to $2 \times 10^6$ cells ml$^{-1}$, and treated with RB at a final concentration of 0.5 μM for 30 min (pretreatment) in light (+) or dark (−). After the pretreatment all the cultures were exposed to an additional 3.75 μM RB (challenge) in low light and collected for measurement of $F_v/F_m$ at 30, 60, and 90 min. The cells were dark-acclimated for at least 30 min before applying a saturating light pulse of 2000 μmol photons m$^{-2}$ s$^{-1}$ and measuring the chlorophyll fluorescence yield using an FMS2 fluorometer (Hansatech Instruments, Norfolk, UK).

## Culture conditions for gene expression analyses by qRT-PCR and RNA-seq

Cultures were grown for at least two light–dark cycles (12 hr light-12 hr dark), and then cell density was adjusted to $2$–$2.5 \times 10^6$ cells ml$^{-1}$ and split into two flasks (one control and the other for RB treatment) at least an hour prior to adding RB to a final concentration of 1 μM. An equal volume of H$_2$O was added to the control. RB was added ~6 hr after the start of the light cycle under light intensity of ~100 μmol photons m$^{-2}$ s$^{-1}$ and the treatment lasted for an hour before harvest. The cells were cooled and harvested by centrifugation at 1200×$g$ for 3 min at 4°C, frozen with liquid nitrogen and stored at −80°C until extraction of RNA. For low light to high light transfer experiment, cultures were grown in continuous light in minimal (HS) medium for 3 days to cell density of $3 \times 10^6$ cells ml$^{-1}$ at 45 μmol photons m$^{-2}$ s$^{-1}$. The light intensity was increased to 500 μmol photons m$^{-2}$ s$^{-1}$ for 1 hr before harvest.

## Gene expression analysis by qRT-PCR

RNA was extracted with TRIzol (Life Technologies, Carlsbad, CA) following manufacturer's instructions and treated with DNaseI (Promega, Madison, WI), then cleaned up using Qiagen RNeasy columns (Qiagen, Germantown, MD). cDNA was synthesized using Omniscript (Qiagen, Germantown, MD) starting with 2–3 μg DNA-free RNA per 20 μl reaction. qPCR was performed using a 7300 FAST qPCR machine (Life Technologies, Carlsbad, CA). The primers were designed with a T$_m$ of 60°C using Primer3 or PrimerExpress (Life Technologies, Carlsbad, CA) (**Table 10**). All primer pairs described in this study were confirmed as having 90–105% amplification efficiency and linear amplification within their dynamic range in experimental samples using serial dilutions of cDNA prior to the experiments. Relative transcript levels were calculated by ΔΔCt method (**Livak and Schmittgen, 2001**) using *CβLP* as internal reference.

## RNA-seq library preparation and analysis

RNA was extracted (**Schmollinger et al., 2014**) and the quality was determined using a 2100 Bioanalyzer (Agilent Technologies, Santa Clara, CA). The triplicate RNA was pooled and 10 μg total

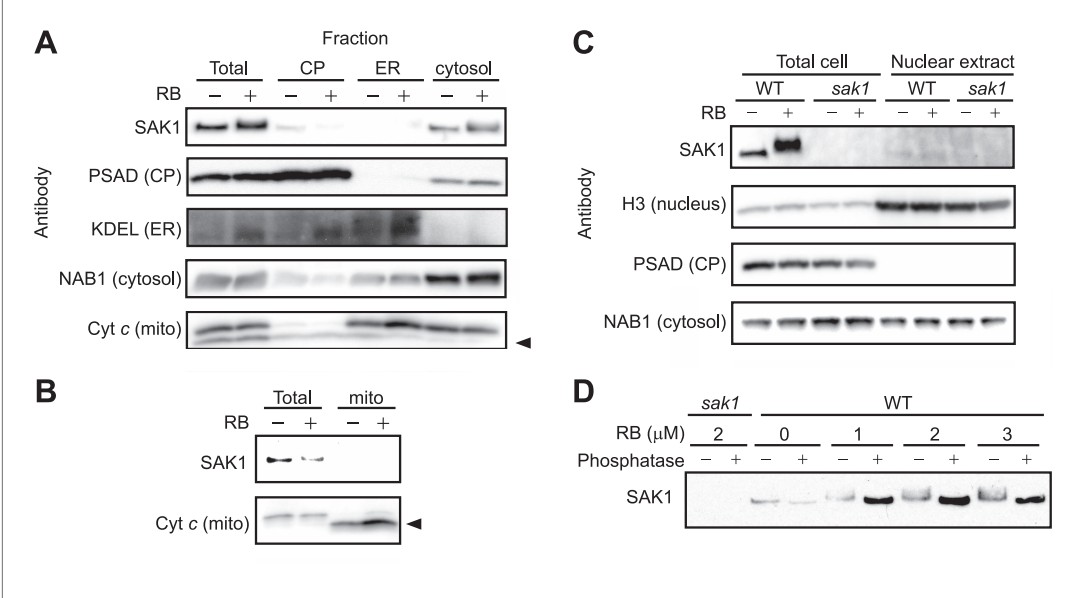

**Figure 6**. SAK1 is a phosphorylated protein that is in the cytosol. (**A** and **B**) SAK1 is detected in the cytosol and not in other subcellular fractions. (**C**) SAK1 is not enriched in nuclear extracts. Approximately 30 µg of protein was loaded into each well except for mitochondrial fractions that were loaded approximately 7.5 µg protein due to low protein yield in isolated fractions. Subcellular markers: Chloroplast (CP), PSAD; Endoplasmic reticulum (ER), KDEL; Cytosol, NAB1; Mitochondria (mito), cytochrome *c* (Cyt c); Nuclear, histone 3 (H3). The arrowhead indicates the band corresponding to Cyt *c*. (**D**) Protein extracts from cells treated with increasing concentrations of RB were then treated with phosphatase (+) or only with buffer (−) before detection of SAK1 by immunoblot analysis.

RNA was used to prepare RNA-seq library according to the manufacturer's protocol (Illumina, San Diego, CA). The quality of the library was assessed using a 2100 Bioanalyzer before sequencing with Genome Analyzer (Illumina, San Diego, CA). Each sample was run in replicates on two lanes. RNA-Seq data was analyzed as before (*Duanmu et al., 2013*). On average, 75% of the sequences could be assigned unambiguously to Augustus v10.2 gene models to generate the matrix of counts per gene. This matrix was used for differential expression analysis using *DESeq* (*Anders and Huber, 2010*) using *per-condition* dispersion estimates and variance stabilization to compute moderate fold changes. Genes were classified as differentially expressed based on a (moderate) twofold regulation and a false discovery rate (FDR) <1%.

## Amplification of cDNA and genomic region of *SAK1* and transformation of *sak1*

Near full-length cDNA was isolated by RT-PCR (described in above section; Gene expression analysis by qRT-PCR) and rapid amplification of cDNA ends (RACE) using GeneRACER (Life Technologies, Carlsbad, CA) as previously described (*Molnar et al., 2009*). Despite multiple attempts the 5′ end of the transcript could not be amplified by 5′-RACE. Because the experimentally obtained CDS differed from the most current v5, it has been deposited to genbank (accession KF985242). Though some differences exist at the nucleotide level, the protein sequence of the resulting CDS was identical to that of au5.g7871_t1. Genomic DNA containing *SAK1* was amplified using primers 5′-CAGGACCGGGCACTGAGTGAAGGTTA-3′ (+) and 5′-ATGATGCACTGTGGGACACGCTGAGT-3′ (−) using PrimeStar HS with GC buffer (Takara/Clontech, Palo Alto, CA) and cloned into pGEM-Teasy after adding an adenine. The resulting plasmid was co-transformed with pBC1 and selected with 1 µM paromomycin. Transformation of *sak1* was performed as described previously (*Kindle et al., 1989*).

## SAK1 antibody generation and protein detection by immunoblotting

To raise antibodies against SAK1, an epitope at the N-terminus of the translated coding sequence of SAK1 (DTLLTPLREDATAESGGDA) was designed, synthesized and injected into rabbits, and the

**Table 9.** Genes up-regulated during both $^1O_2$ acclimation and dark to light transition

| Gene ID (v4) | Gene name | Annotation | RB (log$_2$) | DL (log$_2$) (*Duanmu et al., 2013*) |
|---|---|---|---|---|
| Cre02.g137700.t1.1* | | | 6.49 | 2.34 |
| Cre06.g281250.t1.1* | CFA1 | cyclopropane fatty acid synthase | 5.92 | 4.49 |
| Cre01.g033300.t1.1* | | | 5.72 | 3.62 |
| Cre13.g566850.t1.1* | SOUL2 | SOUL heme-binding protein | 5.53 | 2.25 |
| Cre13.g600650.t1.1* | | | 4.76 | 3.26 |
| Cre06.g263550.t1.1* | LCI7 | R53.5-related protein | 4.46 | 5.27 |
| Cre07.g342100.t1.1* | | | 4.43 | 1.84 |
| Cre09.g398700.t1.1* | CPLD27 | coclaurine N-methyltransferase | 4.05 | 1.36 |
| Cre12.g492650.t1.1* | FAS2 | fasciclin-like protein | 4.01 | 9.24 |
| Cre08.g381510.t1.1* | | | 3.94 | 3.27 |
| Cre10.g458450.t1.2* | GPX5 | glutathione peroxidase | 3.91 | 3.08 |
| Cre11.g474600.t1.1* | | | 3.90 | 1.99 |
| Cre13.g600700.t1.1* | | | 3.78 | 5.79 |
| Cre14.g613950.t1.1* | | | 3.65 | 2.68 |
| Cre06.g269300.t1.1* | | | 3.50 | 1.99 |
| Cre08.g380300.t1.2* | MSRA3 | peptide methionine sulfoxide reductase | 3.45 | 1.79 |
| Cre01.g031650.t1.2* | CGLD12 | protein with potential galactosyl transferase activity | 3.30 | 4.90 |
| Cre14.g629061.t1.1* | | | 3.25 | 1.88 |
| Cre13.g564900.t1.1* | | | 3.22 | 3.38 |
| Cre13.g586450.t1.1 | | | 3.21 | 3.50 |
| Cre02.g139500.t1.1* | | | 3.04 | 2.12 |
| Cre19.g756100.t1.1 | | | 3.04 | 6.53 |
| Cre01.g036000.t1.2 | | | 3.02 | 1.16 |
| Cre14.g618400.t1.1* | | | 2.97 | 2.16 |
| Cre17.g741300.t1.2* | | | 2.88 | 1.92 |
| Cre16.g648700.t1.2* | | | 2.79 | 2.35 |
| Cre17.g729950.t1.1 | | | 2.77 | 2.61 |
| Cre17.g721000.t1.1 | | | 2.70 | 2.12 |
| Cre06.g263500.t1.1* | | | 2.67 | 3.37 |
| Cre01.g016150.t1.1* | | | 2.65 | 2.92 |
| Cre08.g380000.t1.1* | | | 2.59 | 3.74 |
| Cre04.g224800.t1.1 | VAMP74 | R-SNARE protein, VAMP72-family | 2.58 | 3.34 |
| Cre03.g210150.t1.1 | | | 2.57 | 3.44 |
| Cre14.g615600.t1.1* | | | 2.53 | 2.40 |
| Cre06.g293100.t1.1 | | Qc-SNARE SYP6-like protein | 2.48 | 4.90 |
| Cre08.g368950.t1.1 | DHQS | 3-dehydroquinate synthase | 2.39 | 2.49 |
| Cre10.g424350.t1.2 | | metalloprotease | 2.37 | 3.18 |
| Cre12.g537225.t1.1 | | | 2.34 | 3.39 |
| Cre07.g336900.t1.2 | | | 2.32 | 2.31 |
| Cre16.g664050.t1.1 | | | 2.31 | 1.88 |
| Cre16.g677750.t1.1 | | | 2.04 | 2.22 |
| Cre12.g537227.t1.1 | | | 2.00 | 3.46 |

*Table 9. Continued on next page*

*Table 9. Continued*

| Gene ID (v4) | Gene name | Annotation | RB (log$_2$) | DL (log$_2$) (*Duanmu et al., 2013*) |
|---|---|---|---|---|
| Cre17.g737050.t1.1 | | RabGAP/TBC protein | 1.99 | 2.32 |
| Cre06.g297450.t1.1 | | | 1.93 | 1.46 |
| Cre06.g258600.t1.1* | | | 1.91 | 3.63 |
| Cre16.g663950.t1.1 | | SC5D, C-5 sterol desaturase | 1.89 | 2.03 |
| Cre13.g588150.t1.1 | | | 1.86 | 6.21 |
| Cre17.g722150.t1.1 | PKS3 | type III polyketide synthase | 1.85 | 1.61 |
| Cre16.g688550.t1.1 | GSTS1 | glutathione-S-transferase | 1.84 | 1.20 |
| Cre03.g207800.t1.1 | | | 1.84 | 7.09 |
| Cre10.g444550.t1.1* | SPP1A | signal peptide peptidase | 1.81 | 5.33 |
| Cre13.g602500.t1.2 | | | 1.76 | 1.59 |
| Cre03.g163400.t1.2* | | | 1.76 | 2.15 |
| Cre10.g450000.t1.1 | | | 1.74 | 2.18 |
| Cre01.g015500.t1.1 | | | 1.72 | 1.55 |
| Cre02.g105750.t1.2 | | | 1.71 | 3.23 |
| Cre01.g061750.t1.1 | SPT2 | serine palmitoyltransferase | 1.71 | 2.29 |
| Cre83.g796250.t1.1 | | | 1.68 | 1.59 |
| Cre16.g656150.t1.1 | | | 1.67 | 3.55 |
| Cre01.g002050.t1.2 | | | 1.66 | 3.15 |
| Cre12.g556750.t1.2 | Tic32-like 1 | Short-chain dehydrogenase, classical family, similar to PsTic32 | 1.66 | 3.15 |
| Cre12.g559100.t1.1 | | | 1.66 | 3.11 |
| Cre09.g411750.t1.2 | | | 1.61 | 1.96 |
| Cre11.g482650.t1.2 | | | 1.57 | 3.40 |
| Cre06.g310500.t1.1* | | | 1.57 | 6.23 |
| Cre09.g397900.t1.1 | | transmembrane protein | 1.56 | 2.02 |
| Cre04.g215600.t1.1 | | | 1.53 | 2.64 |
| Cre02.g093800.t1.1 | | | 1.51 | 4.99 |
| Cre02.g093750.t1.1 | NRX2 | Nucleoredoxin 2 | 1.50 | 6.26 |
| Cre01.g004350.t1.1 | | | 1.50 | 2.29 |
| Cre01.g034600.t1.1 | | | 1.50 | 2.22 |
| Cre11.g472600.t1.2 | | | 1.48 | 2.00 |
| Cre12.g500500.t1.2 | SMT1 | sterol-C24-methyltransferase | 1.46 | 3.05 |
| Cre13.g577950.t1.1 | VPS6 | subunit of the ESCRT-III complex | 1.45 | 2.36 |
| Cre02.g118200.t1.1 | | | 1.44 | 2.79 |
| Cre01.g012500.t1.1 | PRA1 | prenylated rab acceptor family protein | 1.43 | 2.46 |
| Cre12.g521600.t1.2 | | | 1.42 | 2.89 |
| Cre03.g179100.t1.1 | | ubiquitin fusion degradation protein | 1.41 | 3.38 |
| Cre09.g413150.t1.2 | | | 1.39 | 4.31 |
| Cre13.g572200.t1.1 | | tyrosine/tryptophan transporter protein | 1.39 | 2.57 |
| Cre03.g185850.t1.2 | | PfkB-type carbohydrate kinase | 1.37 | 3.05 |
| Cre18.g743600.t1.1 | | | 1.37 | 1.65 |
| Cre02.g076800.t1.1 | | sterol reductase | 1.36 | 2.41 |
| Cre06.g256750.t1.1 | FAT1 | acyl carrier protein thioesterase | 1.35 | 1.67 |

*Table 9. Continued*

| Gene ID (v4) | Gene name | Annotation | RB (log$_2$) | DL (log$_2$) (*Duanmu et al., 2013*) |
|---|---|---|---|---|
| Cre17.g729450.t1.1 | | | 1.34 | 1.90 |
| Cre11.g471550.t1.1 | | | 1.34 | 3.29 |
| Cre09.g395750.t1.2 | | | 1.33 | 2.87 |
| Cre14.g617100.t1.1 | | | 1.33 | 3.33 |
| Cre16.g691500.t1.1 | | Sec14p-like lipid-binding protein | 1.33 | 2.28 |
| Cre02.g079550.t1.1 | DRP2 | Dynamin-related GTPase | 1.32 | 2.34 |
| Cre02.g079300.t1.1 | VPS4 | AAA-ATPase of VPS4/SKD1 family | 1.32 | 1.96 |
| Cre05.g231700.t1.2 | | | 1.31 | 2.40 |
| Cre02.g132300.t1.2 | DNJ12 | DnaJ-like protein | 1.30 | 2.24 |
| Cre69.g794101.t1.1 | | | 1.30 | 2.65 |
| Cre13.g565600.t1.2 | | | 1.29 | 3.42 |
| Cre13.g593700.t1.1 | | monooxygenase, DBH-like | 1.29 | 1.81 |
| Cre12.g498000.t1.2 | | | 1.28 | 3.88 |
| Cre06.g292900.t1.2 | | | 1.28 | 2.16 |
| Cre08.g372100.t1.1 | HSP70A | Heat shock protein 7A | 1.27 | 2.28 |
| Cre01.g039350.t1.1 | NCR2 | NADPH-cytochrome P45 reductase | 1.26 | 2.19 |
| Cre03.g211100.t1.1 | | | 1.26 | 2.11 |
| Cre17.g731800.t1.1 | | | 1.25 | 1.78 |
| Cre17.g730650.t1.1 | | | 1.25 | 2.28 |
| Cre02.g123000.t1.2 | | | 1.24 | 1.42 |
| Cre05.g247700.t1.2 | | | 1.24 | 2.71 |
| Cre08.g360800.t1.2 | | haloacid dehalogenase-like hydrolase | 1.23 | 4.39 |
| Cre07.g350750.t1.1 | PTOX1 | alternative oxidase | 1.22 | 3.32 |
| Cre17.g703750.t1.1 | | | 1.20 | 2.21 |
| Cre06.g306041.t1.1 | | | 1.20 | 2.90 |
| Cre02.g116650.t1.1 | | | 1.20 | 2.83 |
| Cre08.g379400.t1.2 | | | 1.18 | 3.04 |
| Cre16.g677000.t1.1 | HSP70E | Heat shock protein 7E | 1.18 | 2.50 |
| Cre06.g283900.t1.1 | | | 1.18 | 5.24 |
| Cre14.g626750.t1.1 | | | 1.17 | 4.12 |
| Cre01.g010700.t1.1 | | | 1.16 | 2.10 |
| Cre01.g002000.t1.2 | | predicted proteim | 1.15 | 1.68 |
| Cre04.g213150.t1.1 | | | 1.15 | 2.78 |
| Cre16.g694250.t1.1 | | | 1.15 | 2.92 |
| Cre05.g246400.t1.1 | | | 1.15 | 2.74 |
| Cre02.g128450.t1.1 | | | 1.13 | 2.82 |
| Cre03.g180250.t1.1 | | Myo-inositol-1-phosphate synthase | 1.13 | 2.05 |
| Cre03.g186150.t1.1 | | | 1.13 | 1.78 |
| Cre02.g137800.t1.1 | | | 1.13 | 2.00 |
| Cre11.g471500.t1.1 | MFT10 | predicted protein | 1.11 | 1.40 |
| Cre10.g435200.t1.1 | | | 1.10 | 2.13 |
| Cre13.g593850.t1.2 | | | 1.10 | 3.91 |

*Table 9. Continued*

| Gene ID (v4) | Gene name | Annotation | RB (log$_2$) | DL (log$_2$) (*Duanmu et al., 2013*) |
|---|---|---|---|---|
| Cre19.g754000.t1.2 | | | 1.10 | 2.33 |
| Cre13.g593869.t1.1 | | | 1.10 | 3.90 |
| Cre08.g377300.t1.2 | | | 1.09 | 3.27 |
| Cre04.g225050.t1.2 | | predicted protein | 1.09 | 3.55 |
| Cre07.g330300.t1.1 | | | 1.08 | 2.22 |
| Cre12.g500450.t1.2 | | | 1.08 | 3.00 |
| Cre06.g262000.t1.1 | | | 1.08 | 1.87 |
| Cre10.g441550.t1.2 | *MAM3B* | predicted protein | 1.07 | 1.54 |
| Cre06.g249800.t1.1 | | unknown conserved protein | 1.07 | 2.08 |
| Cre01.g038250.t1.1 | *SDC1* | serine decarboxylase | 1.06 | 1.92 |
| Cre44.g788200.t1.1 | | | 1.06 | 2.13 |
| Cre08.g359200.t1.2 | | | 1.03 | 2.69 |
| Cre05.g245950.t1.1 | *DRP1* | Dynamin-related GTPase | 1.03 | 2.15 |
| Cre05.g234100.t1.1 | *CYP745A1* | cytochrome P45 | 1.01 | 2.61 |
| Cre07.g328700.t1.2 | | | 1.01 | 1.56 |
| Cre10.g440250.t1.2 | | | 1.01 | 2.14 |
| Cre17.g725200.t1.1 | | MDR-like ABC transporter | 1.01 | 3.30 |
| Cre82.g796100.t1.1 | | | 1.01 | 2.49 |

*Genes defined as SAK1-dependent in **Table 4**.

resulting crude serum was affinity purified (Open Biosystems/Thermo Scientific, Waltham, MA). For immunoblot detection of SAK1, proteins were separated with NuPAGE 3–8% Tris Acetate gels (Life Technologies, Carlsbad, CA) and transferred to nitrocellulose membranes. All other blots were prepared from running the protein on 10–20% Tris-glycine gels and transferring to a PVDF membrane. The membranes were blocked for several hours in 5% milk in TBS-T, incubated with the primary antibody overnight, then with secondary antibody for several hours in 1% milk TBS-T before washing and developing with a chemiluminescence detection kit. Commercial antibodies were anti-histone H3 (ab1791; Abcam, Cambridge, UK) and anti-KDEL (ab12223; Abcam, Cambridge, UK). Other antibodies were generous gifts from Jean-David Rochaix (anti-PSAD), Olaf Kruse (anti-NAB1), and Patrice Hamel (anti-cytochrome *c*).

## Subcellular fractionation and protein quantification

Nuclear fractions were prepared from 450 ml of synchronized cultures with ~2 × 10$^6$ cells ml$^{-1}$ that had been incubated with or without 2 µM RB under light for 40 min. The cells were collected and treated with autolysin for 40 min and examined for the removal of cell walls by addition of 1 volume of 0.1% Triton-X. Nuclear extract was prepared as described previously (*Winck et al., 2011*) using CelLytic PN kit (Sigma-Aldrich, St. Louis, MO). Because there were bands detected in the nuclear extract close to the size of SAK1, nuclear extract was prepared from WT (4A+) and *sak1* rather than a cell wall-deficient strain (*cw15*). Chloroplasts were isolated from cell wall-less strain *cw15* as described previously (*Klein et al., 1983*). Mitochondria were isolated as described (*Eriksson et al., 1995*). After unbroken cells, chloroplasts, and mitochondria were collected, the ER fraction was collected by centrifugation at 100,000×*g* for 90 min at 4°C. The remaining supernatant was enriched for cytosol. Protein was extracted and prepared for SDS-PAGE as described (*Calderon et al., 2013*) with minor modifications. Protein was quantified by using BCA1 kit (Sigma-Aldrich, St. Louis, MO) after extraction with the methanol-chloroform method (*Wessel and Flügge, 1984*).

**Table 10.** Primers used for qRT-PCR analyses

| v4 ID | v5 ID | Gene name | Forward | Reverse |
|---|---|---|---|---|
| Cre01.g007300.t1.1 | Cre01.g007300.t1.2 | | AGCATGTGCGTGTGGAGTAG | CCTTACCATAGGCCTGACCA |
| au5.g10700_t1a | Cre03.g177600.t1.3 | | CTGGACATGTCGGCTATGAA | GCTCATGTCGTACTCCAGCA |
| au5.g13389_t1* | Cre06.g299700.t1 | SOUL1† | TGCGTATGGGTGTCCACTAA | TGGGGATCTTCTTCATGTCC |
| Cre06.g263550.t1.1 | Cre06.g263550.t1.2 | LCI7 | TTTGGTTGCGTTGCATGTAT | TCAACGCGGTGTCAAACTTA |
| Cre06.g281250.t1.1 | Cre06.g281250.t1.2 | CFA1 | CCTACAACGACAACGACGTG | GGAAGTTCCAGGATGACCAG |
| Cre06.g298750.t1.1 | Cre06.g298750.t1.2 | AOT4 | CCGTGTGCACAGATTCAAAG | CACACAGCGCCTCCTACATA |
| Cre08.g358200.t1.2 | Cre08.g358200.t2.1 | | TGTGGCATCAAGGTGTGTTGT | AACCCCACACCCCTCTCTTT |
| Cre09.g398700.t1.1 | Cre09.g398700.t1.2 | CFA2 | CGACCTGCTGCTCACTTCC | GTGTAGGCGGTGGTCAAGAT |
| Cre10.g458450.t1.2 | Cre10.g458450.t1.3 | GPX5 | AACCAATCGCCTAACACCTG | CACTTGCTAGCCACGTTCAC |
| Cre12.g503950.t1.1 | Cre12.g503950.t1.2 | | GGAGGGAGTACCACGAGACA | GATTGCTGTAAGGCCGGATA |
| Cre13.g564900.t1.1 | Cre13.g564900.t1.2 | MRP3 | TCATGACGTACATCTCGATTCTCA | AGGGAATGTAGTAGCGCTGAATG |
| au5.g4402_t1* | Cre13.g566800.t1.2 | | TGCTTGGAAGACCCACTTTT | GAGCTGGAGTTGCAGTTGTG |
| Cre13.g566850.t1.1 | Cre13.g566850.t1.2 | SOUL2 | CCCTCCCCTCCTTCAGACTA | CGTACCTGAGCGCGCATATTT |
| Cre14.g613950.t1.1 | Cre14.g613950.t2.1 | | CGCCAACCCCATGATC | CCGCAACGTACCGTGATG |
| Cre16.g683400.t1.1 | Cre16.g683400.t1.2 | | CCTGAACAAACACACGATGG | GAACGCCGTCAAATCATCTT |
| Cre16.g688550.t1.1 | Cre16.g688550.t1.2 | GST1 | AGTGCGGAGGAAGTCGTAAA | GTAAAAGACGTGCGTGCAAA |
| g6364.t1 | | CβLP(RCK1) | GAGTCCAACTACGGCTACGC | GGTGTTCAGGTCCCACAGAC |
| Cre14.g623650.t1.1 | Cre14.g623650.t1 | | GACAACGCGGCCTACAAGA | CCGAGCTGGCGGTGTTAA |
| au5.g2281_t1* | g16723.t1 | MKS1 | GCTTGAGCGCGAGACGAA | CGCTGAAAGCATTGCAGAAG |
| Cre08.g380300.t1.2 | Cre08.g380300.t1.2 | | ACCACCAGCAGTACCTGTCC | CGCTCCAATAAAGCCTTCAG |
| au5.g7871_t1‡ | (Cre17.g741300.t1.2)‡ | SAK1(5'UTR) | CAAGTGCTCATGAGAGGCCTTA | TACGTCATCCAGTTCCACATCC |
| au5.g7871_t1‡ | (Cre17.g741300.t1.2)‡ | SAK1(3'UTR) | TCAAGCGTGTGGGTAAGAGCTA | ACGCTATCTCCGTCCTAATCCA |
| Cre08.g365900.t1.1 | Cre08.g365900.t1.2 | LHCSR1 | CACACAATTCTGCCAACAGC | ATCTGCTTCACGGTTTGGTC |
| Cre04.g220850.t1.1 | Cre04.g220850.t1.2 | | TAATGGTATGGATGCGGTCA | ACTGCCAGTTATGGGTCCTG |
| Cre09.g395750.t1.2 | Cre09.g395750.t1.3 | | ACCGTCCGTGAACCTTACTG | CGCAAACACGTCTCAAAGAA |

*Was originally mapped and identified as augustus version 5 models within *Chlamydomonas* genome v4.

†*SOUL1* was given the name in v4 but not v5.

‡Primers were designed against experimentally obtained cDNA (Genbank accession KF985242) and differs from v5. Closest gene model is au5.g7871_t1.

## Acknowledgements

We would like to thank Deqiang Duanmu and Cinzia Formighieri for discussions on subcellular fractionation, David Lopez, Ian Blaby, and Simon Prochnik for guidance on functional analysis of RNA-seq data and gene ID identification, Attila Molnar for advice on RACE, and Olaf Kruse, Patrice Hamel, and Jean-David Rochaix for gifts of antibodies. This project was supported by Award Number R01GM071908 from the National Institute of General Medical Sciences and by the Howard Hughes Medical Institute and the Gordon and Betty Moore Foundation (through Grant GBMF3070) to KKN and National Institutes of Health R24 GM092473 for RNA Seq data analysis. The content is solely the responsibility of the authors and does not necessarily reflect the official views of the National Institute of General Medical Sciences or the National Institutes of Health.

## Additional information

### Funding

| Funder | Grant reference number | Author |
|---|---|---|
| Howard Hughes Medical Institute | | Krishna K Niyogi |
| Gordon and Betty Moore Foundation | GBMF3070 | Krishna K Niyogi |
| National Institutes of Health | R24 GM092473 | Matteo Pellegrini, Sabeeha S Merchant |
| National Institute of General Medical Sciences | R01 GM071908 | Krishna K Niyogi |

The funders had no role in study design, data collection and interpretation, or the decision to submit the work for publication.

### Author contributions

SW, Conception and design, Acquisition of data, Analysis and interpretation of data, Drafting or revising the article; BLC, RMD, Acquisition of data, Drafting or revising the article; HKL, Acquisition of data; DC, Acquisition of data, Analysis and interpretation of data, Drafting or revising the article; MP, Acquisition of data, Analysis and interpretation of data; SSM, Analysis and interpretation of data, Drafting or revising the article; KKN, Conception and design, Analysis and interpretation of data, Drafting or revising the article

## Additional files

### Supplementary file

• Supplementary file 1. Genes that display significant differential expression by pair-wise comparisons.

### Major dataset

The following dataset was generated:

| Author(s) | Year | Dataset title | Dataset ID and/or URL | Database, license, and accessibility information |
|---|---|---|---|---|
| Wakao S, Chin BL, Ledford HK, Dent RM, Casero D, Pellegrini M, Merchant SS, Niyogi KK | 2014 | Data from: Phosphoprotein SAK1 is a regulator of acclimation to singlet oxygen in Chlamydomonas reinhardtii | http://dx.doi.org/doi:10.5061/dryad.h7pm2 | Available at Dryad Digital Repository under a CC0 Public Domain Dedication. |

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
