## [Decision Letter]

Thank you for sending your work entitled “Phosphoprotein SAK1 is a regulator of acclimation to singlet oxygen in *Chlamydomonas reinhardtii*” for consideration at *eLife*. Your article has been favorably evaluated by a Senior editor, Detlef Weigel, and 2 reviewers, one of whom, Todd Mockler, is a member of our Board of Reviewing Editors.

The Reviewing editor and the other reviewer discussed their comments before we reached this decision, and the Reviewing editor has assembled the following comments to help you prepare a revised submission.

This manuscript from Wakao et al addresses SAK1, a cytoplasmic phosphoprotein that is a component of the retrograde signaling pathway between the plastid and nucleus. SAK1 functions in regulation of nuclear gene expression during acclimation of Chlamydomonas cells to reactive oxygen species (ROS). This is an interesting and timely manuscript on an interesting topic. While ROS are implicated in pathologies including stresses, signal transduction, and developmental regulation in multiple systems, ROS signaling pathways remain to be elucidated in detail. Singlet oxygen (1O2) is a particularly toxic form of ROS that is generated during photosynthesis. Plants and algae have evolved retrograde signaling between the chloroplast and the nucleus through which singlet oxygen species modulate nuclear gene expression and acclimation to oxidative stress. Specifically relevant to this manuscript, few components of singlet oxygen signaling have been discovered in any system, and there remains a large gap between singlet oxygen generation in the chloroplast and mediation of gene expression regulation in the nucleus.

This work concerns the characterization of a *Chlamydomonas* mutant called *sak1* that is unable to increase its tolerance to singlet oxygen. SAK1 is a novel cytosolic phosphoprotein containing a domain that is conserved among some bZIP transcription factors. SAK1 accumulation is itself induced and phosphorylated upon exposure to singlet oxygen, consistent with its function in the retrograde signal transduction pathway(s) leading to acclimation to singlet oxygen stress. Gene responses to RB are shown to be substantially affected in the *sak1* mutant. The protein encoded by SAK1 is shown to be located in the cytosol and phosphorylated upon exposure of the microalgal cells to RB. The experiments seem to have been done carefully and the manuscript is written in a clear and concise manner.

In summary, this is a well-written manuscript that presents a new protein, identified in a clever mutant screen that is implicated in both retrograde signaling and ROS response. This factor is of general interest due to its relevance to photosynthesis and primary metabolism, retrograde signaling, ROS, and abiotic stress responses.

Specific concerns to be addressed in the revised manuscript and response to the reviews are as follows:

1) Are the data presented here really representative of the response of photosynthetic organisms to natural conditions of 1O2 stress, e.g. excess light energy? In Figure 1, it is shown that RB-pretreated WT cells become resistant to high concentrations of RB. However, it is not shown whether or not RB-treated cells are also resistant to high light or Norflurazon, two conditions known to produce 1O2 in the chloroplasts. These data should be shown in the manuscript to demonstrate that the acclimation mechanism described here can be extrapolated to natural 1O2 stress conditions.

2) Did the authors compare the RB-induced changes in gene expression with the gene expression profile in high light-exposed Chlamydomonas cells?

3) Intriguingly, the *sak1* mutant grows in high light similarly to the WT strain, and it is not more sensitive to Norflurazon compared to WT (Figure 1). This suggests that the sak1 mutation does not affect the response to 1O2 when it is produced in the chloroplast. Is this absence of phenotype of the *sak1* mutant also observed when the HL and Norflurazon treatments were imposed after acclimation to RB?

4) During acclimation of WT to RB, genes involved in transport constitute one of the most enriched classes (Table 3). The authors interpret this striking effect as a response of the cells to pump RB out, reinforcing the idea that at least part of the changes in gene expression is a response to RB itself rather than a response to singlet oxygen. In this context, it would be useful to compare the expression of the genes shown in Figure 2 when RB is imposed in the dark (no 1O2 produced) or in the light.

5) The authors should pay more attention to the physiological meaning of the treatments that they used to isolate and characterize their mutant. A few additional experiments could be helpful to convince the readers that the gene responses described in this manuscript can be useful to understand the responses of photosynthetic organisms to 1O2 produced in vivo under excess light energy.

6) Are Figure 2 and Table 1 redundant? They seem to present the same data and of the two, Table 1 is more complete because it presents both the RNA-seq and qRT-PCR results. The standard deviations could be added to Table 1 and would make Figure 2 unnecessary.

---

## [Author Response]

*1) Are the data presented here really representative of the response of photosynthetic organisms to natural conditions of 1O2 stress, e.g. excess light energy? In*
Figure 1*, it is shown that RB-pretreated WT cells become resistant to high concentrations of RB. However, it is not shown whether or not RB-treated cells are also resistant to high light or Norflurazon, two conditions known to produce 1O2 in the chloroplasts. These data should be shown in the manuscript to demonstrate that the acclimation mechanism described here can be extrapolated to natural 1O2 stress conditions*.

We have published previously that WT Chlamydomonas can acclimate to RB following pretreatment with high light (35), indicating that 1O2 generation by excess light elicits a response overlapping with the response to RB. We have repeated this experiment including sak1, and the results in new Figure 1 show that the mutant acclimates less effectively. The mutant does exhibit some acclimation, however, which we attribute to the fact that high light initiates several signaling pathways in addition to 1O2 (e.g. reduced PQ pool, H2O2, O2-), to which sak1 is still capable of responding, and we have previously shown that a more general oxidative stress response can induce resistance to RB (21).

On the other hand, pretreatment with RB did not increase resistance to HL or norflurazon in WT or sak1 (new Figure 1—figure supplement 1). Acclimation to 1O2 is transient, lasting less than 24 hours (35), and SAK1 appears to be involved in the early and transient response to 1O2 during perturbations rather than actively detoxifying during long-term, chronic stresses, as can be seen in the wild-type growth of sak1 on medium containing norflurazon or in HL (Figure 1 in the revised Figure 1). We speculate that over the days required to see growth in HL or norflurazon, the cells are able to adjust their physiology in different ways to reduce 1O2 generation and/or toxicity.

*2) Did the authors compare the RB-induced changes in gene expression with the gene expression profile in high light-exposed* Chlamydomonas *cells?*

We have tested the expression of several of the 1O2-responsive genes (new Table 1) and the SAK1 transcript (new Figure 4) during a low to high light transition and found that they are moderately induced compared to induction by RB. This was expected because 1O2 is a part of the high light response, whereas RB treatment isolates and amplifies the 1O2-dependent part of this response.

*3) Intriguingly, the sak1 mutant grows in high light similarly to the WT strain, and it is not more sensitive to Norflurazon compared to WT (*Figure 1*). This suggests that the* sak1 *mutation does not affect the response to 1O2 when it is produced in the chloroplast*. *Is this absence of phenotype of the sak1 mutant also observed when the HL and Norflurazon treatments were imposed after acclimation to RB?*

Please see our responses to points 1 and 2 above. High light is known to induce 1O2 production in the chloroplast, so our observation of impaired RB acclimation in sak1 after pretreatment with high light (new Figure 1) suggests that sak1 does indeed affect the response to 1O2 produced in the chloroplast. As described above, the acclimation induced by RB pretreatment is transient (35) and thus presumably is not maintained long enough to confer resistance to chronic high light or norflurazon stress.

*4) During acclimation of WT to RB, genes involved in transport constitute one of the most enriched classes (*Table 3*). The authors interpret this striking effect as a response of the cells to pump RB out, reinforcing the idea that at least part of the changes in gene expression is a response to RB itself rather than a response to singlet oxygen. In this context, it would be useful to compare the expression of the genes shown in*
Figure 2
*when RB is imposed in the dark (no 1O2 produced) or in the light*.

We examined this issue by performing an additional experiment comparing gene expression in the dark with or without RB (new Table 4), and we found that the expression levels of the 1O2 target genes (including specific ABC transporter genes) remained unchanged when cells were exposed to RB in the dark. This result demonstrates that the induction of these genes is due to 1O2 rather than RB itself. The RB-dark vs. RB-light comparison strongly resembled that of mock-light vs. RB-light and did not directly address whether the some of the genes were responding to the chemical RB or 1O2. Besides being produced by excess light, 1O2 is known to be generated by naturally occurring photosensitizers (e.g. humic substances) in aquatic and terrestrial environments. Microbes and phytoplankton must respond to this stress, and our results are relevant to understanding how this occurs. We have added text and references to clarify and discuss these topics.

*5) The authors should pay more attention to the physiological meaning of the treatments that they used to isolate and characterize their mutant. A few additional experiments could be helpful to convince the readers that the gene responses described in this manuscript can be useful to understand the responses of photosynthetic organisms to 1O2 produced in vivo under excess light energy*.

In addition to the cross-acclimation experiments (new Figure 1 and new Figure 1—figure supplement 1), we performed an experiment to measure Fv/Fm (maximum efficiency of photosystem II) during RB treatments (new Figure 1). The data show that 1O2 generated during RB treatment causes inhibition of photosystem II in the chloroplast and that pretreatment alleviates this inhibition, consistent with the growth phenotype seen in Figure 1.

*6) Are*
Figure 2
*and*
Table 1
*redundant? They seem to present the same data and of the two,*
Table 1
*is more complete because it presents both the RNA-seq and qRT-PCR results. The standard deviations could be added to*
Table 1
*and would make*
Figure 2
*unnecessary*.

As described in the Material and methods, our RNA-seq was performed with a single, pooled RNA-seq library prepared from biological triplicates. Therefore we feel it is important to validate the expression profiles of at least some of the genes. Figure 2 has extra information compared to the Table on the basal level of expression in the mutant compared to WT rather than only fold change; some genes are already elevated in the sak1 mutant without RB, suggesting that the mutant is experiencing some constitutive stress. We therefore would like to keep Figure 2 as is and have moved the original Table 1 that compares fold changes calculated from RNA-seq and qPCR to accompany Figure 2 and renamed it Figure 2.